# The Impact of Varieties and Growth Stages on the Production Performance and Nutritional Quality of Forage Triticale in the Qaidam Basin

**DOI:** 10.3390/plants14192942

**Published:** 2025-09-23

**Authors:** Fengjuan Xue, Chengti Xu, Yuanyuan Zhao, Xiaojian Pu, Jie Wang, Xiaoli Wei, Wei Wang

**Affiliations:** 1Academy of Animal Science and Veterinary Medicine, Qinghai University, Xining 810016, China; ys230951310641@qhu.edu.cn (F.X.); 18893147262@163.com (Y.Z.); pu_x_j@163.com (X.P.); wangjie08142023@163.com (J.W.); 18894499178@163.com (X.W.); 2Key Laboratory of Northwest Cultivated Land Conservation and Marginal Land Improvement Enterprises, Ministry of Agriculture and Rural Affairs, Delingha 817000, China

**Keywords:** triticale, Qaidam Basin Yield, nutritional quality, growth period, TOPSIS methods

## Abstract

Insufficient forage is a significant factor limiting the development of animal husbandry in high-altitude pastoral areas. This study aims to identify suitable high-quality forage Triticale (x *Triticosecale* Wittmack) varieties for cultivation in the Qaidam Basin and determine their optimal harvest period. Seven triticale varieties were selected as experimental materials, and their production performance and nutritional quality were comprehensively analyzed at four different growth stages [booting stage (BTS), heading stage (HDS), flowering stage (FLS), and milk stage (MKS)] in 2024 and 2025, utilizing Pearson correlation analysis, PCA, and TOPSIS methods. The results indicated that as the reproductive period progressed, plant height, dry matter yield, and dry matter content exhibited a continuous increase. In contrast, indicators such as stem diameter, number of green leaves, and fresh biomass yield initially increased before subsequently declining. The MKS soluble sugar content (SS) and relative feed value (RFV) were the highest, crude protein (CP) and crude ash content (CA) decreased, the neutral detergent fiber content (NDF) and the acid detergent fiber (ADF) content reached their peak in the HDS. The relative forage quality (RFQ) reached its peak during the flowering period. Comprehensive analysis showed that the top five rankings are QSM-8 > JSM-3 > QSM-1 > JSM-2 > QSM-7. In the Qaidam Basin, the optimal harvest period from the FLS to the MKS ensures both high yield and good nutritional quality, making it suitable for promotion in the Qaidam Basin region and similar ecological zones.

## 1. Introduction

The Qaidam Basin is situated in the northwest of Qinghai Province, China, on the northeastern edge of the Tibetan Plateau [1]. Owing to its high-altitude topography as a typical inland plateau basin, it is characterized by a plateau continental climate [2]. This region is characterized by an arid and frigid climatic regime, dry and rarefied air, pronounced diurnal temperature fluctuations between morning and evening, and intense solar radiation—with an average solar radiation flux exceeding 6000 joules per square meter per day [3,4]. Notably, it is distinguished by its high altitude and salinity [5]. Despite the availability of abundant light and heat resources, the quality of natural pastures is generally poor, primarily due to frequent frost, wind erosion, and water scarcity, which contribute to severe pasture degradation. As a major saline-alkali area in western China, the Qaidam Basin suffers from severe soil salinization. Meanwhile, forage scarcity has become a critical constraint on the development of agriculture and animal husbandry in northwestern China. With the rapid growth of the livestock industry in this area, the carrying capacity of natural pastures has become inadequate to meet the increasing demands of livestock production, leading to an increasingly prominent contradiction between forage supply and demand. In recent years, the proportion of artificial forage cultivation in the Qaidam Basin has been severely insufficient, thus hindering the sustainable development of the local animal husbandry industry. Therefore, establishing an artificial grass planting system is a crucial strategy to alleviate the forage supply–demand imbalance in the Qaidam Basin.

Triticale (x *Triticosecale* Wittmack), recognized as a dual-purpose crop for both grain and forage production, is distinguished by its high biomass yield, salt–alkali tolerance, broad adaptability, and superior forage quality. These characteristics render it a promising candidate for cultivation in saline-alkali soils and high-altitude regions. Triticale is a novel species formed through the sexual hybridization of wheat (*Triticum*) and rye (*Secale*), followed by chromosome doubling [6]. This crop exhibits pronounced heterosis, effectively combining the high yield and quality traits of wheat with the drought and cold tolerance of rye, thus establishing itself as a high-quality forage option [7,8]. Triticale is abundant in crude protein, soluble sugars, vitamins, and beneficial fibers, and is utilized globally as a premium forage source [9]. It is extensively cultivated in developed livestock countries, the pastoral regions of China, and farming–pastoral transitional zones [10]. Currently, in the farming–pastoral transitional zone of Qinghai, triticale demonstrates superior yield among annual forage types, with its planting area increasing annually. It plays a crucial role in addressing the shortage of high-quality forage in alpine regions and effectively mitigating the hazards of sandstorms, showcasing extensive application prospects.

In recent years, extensive research conducted by scholars worldwide has focused on evaluating the production performance and nutritional quality of triticale across diverse ecological zones, with the explicit objective of identifying elite varieties optimized for cultivation in cold and arid environments. However, due to the ecological constraints of the Qaidam Basin, systematic research on the adaptability evaluation of triticale in this region has not yet been conducted. Furthermore, the dynamic synergistic relationships between different varieties and growth stages of triticale concerning yield and quality remain unclear. Currently, the determination of the optimal harvesting period is predominantly based on empirical experience and lacks robust scientific data. This study aims to evaluate and screen various triticale varieties in the Qaidam Basin to identify those suitable for local cultivation. Consequently, the present research is designed to test the hypothesis that significant differences exist between cultivars and across growth stages regarding yield components and quality traits. The primary research objective is to systematically evaluate the interactions between cultivar-specific traits and phenological development, and how these interactions influence specific morphological traits and yield metrics, such as plant height, fresh biomass yield, dry matter yield, and quality parameters, including nutrient content and forage value. The research findings will enhance our understanding of the growth characteristics and quality traits of triticale at various growth stages in the Qaidam Basin. This understanding will provide a scientific foundation for establishing an effective cultivation technology system for triticale, determining the optimal harvesting period, ensuring the supply of high-quality forage in the region, and promoting the balanced development of grass and livestock.

## 2. Results

### 2.1. Phenological Stages of Different Triticale Varieties

This study observed and compared the key growth stages (booting stage, heading stage, flowering stage, and milking stage) of seven triticale varieties across two years, 2024 and 2025 (Appendix A). The results indicated significant variations in phenological stages among the different varieties. In 2024, QSM-7 and QSM-8 exhibited the longest booting periods, lasting 11 days, whereas QSM-1 had the shortest, lasting only 8 days. With the exception of QSM-3 and JSM-3, the booting periods of the remaining varieties remained relatively stable in 2025. In 2024, QSM-7 reached anthesis the earliest, while QSM-1, QSM-3, and JSM-3 developed more slowly. The anthesis periods of most varieties were longer in 2024 than in 2025. In 2024, QSM-7 and JSM-2 had the longest flowering periods, lasting 16 days, whereas in 2025, the average flowering duration was 14 days. QSM-1, QSM-2, and QSM-3 had flowering periods of only 11 days. In 2024, QSM-3 had the longest ripening period, lasting up to 22 days, while in 2025, there were no significant differences in ripening periods among the varieties.

### 2.2. The Effects of Different Varieties and Growth Stages on the Agronomic Traits and Yield of Feeding Triticale

The two-way ANOVA analysis showed (Table 1) that both variety and growth periods significantly affected the production performance of triticale (*p* < 0.05); the growth stage had a highly significant effect on plant height, tiller number, number of leaves, fresh biomass yield, dry matter yield, and dry matter content, but no significant effect on stem diameter in 2024 (*p* > 0.05); The interaction between the variety and growth stage had a highly significant effect on plant height and dry matter yield (*p* < 0.05), but no significant effect on tiller number, stem diameter, green leaf number, fresh biomass yield, and dry matter content in 2024 (*p* > 0.05). In 2025, varieties had extremely significant effects on plant height, stem diameter, number of leaves, fresh biomass yield, and dry matter yield (*p* < 0.05), with no significant difference in tiller number (Table 2). The growth stage showed no significant difference in tiller number and stem diameter (*p* > 0.05). The interaction between the variety and growth stage led to significant differences in plant height, fresh biomass yield, and dry matter yield, but no significant difference in other indicators (*p* > 0.05).

The agronomic traits of the seven tested triticale varieties at different growth stages showed that, as the growth period progressed, the plant height of triticale exhibited a continuous increasing trend overall in 2024 (Figure 1A), with a decelerated growth rate during the milk stage, reaching the maximum average plant height. Consistent with the growth trends observed in 2025 (Figure 2A), the plant height at the milk stage in 2024 was 24.99% higher than that at the booting stage, and in 2025, this increase reached 25.35%. In 2024, the plant height of variety JSM-3 was significantly higher than that of other varieties across all four growth stages, with an average height of 148.61 cm (*p* < 0.05). There was no significant difference between JSM-2 and QSM-7 at the flowering stage and the milk maturity stage (*p* > 0.05). In 2025, JSM-3 remained the tallest at the milk-ripe stage, reaching a maximum height of 150.23 cm. However, at the flowering stage, JSM-2 exhibited the tallest height, measuring 136.27 cm, and there was no significant difference compared to QSM-7 (*p* > 0.05). The tillering number of each variety was the highest at the booting stage and then significantly decreased as the growth period progressed (Figure 1B), and there was no significant difference among the varieties in both 2024 and 2025 (*p* > 0.05). Variety JSM-3 exhibited the highest number of tillers, averaging 5.60 and 5.13 tillers per plant in 2024 and 2025, showing no significant difference from QSM-7, which averaged 5.53 tillers per plant (Figure 1B). The stem diameter peaked at the heading stage (Figure 1C), with an average stem diameter of 4.52 mm, decreasing by 7.36% at the milk stage. Variety JSM-2 had the highest stem diameter at the heading stage, averaging 4.86 mm, while QSM-1 exhibited an overall lower stem diameter, averaging 3.60 mm. In 2025, the average stem diameter during the booting stage was the highest, while the average stem diameter during the flowering stage was the lowest (Figure 2C). The stem diameter of JSM-3 was significantly higher than that of other varieties, reaching 4.94 mm. The number of green leaves of the tested varieties overall showed a significant trend of increase followed by decrease (Figure 1D), peaking at the flowering stage. There was no significant difference in the number of green leaves among all varieties (*p* > 0.05). This trend was consistent with the growth observed in 2025 (Figure 2D). Compared to the booting stage, the number of green leaves increased by 15.10% in 2024. The varieties JSM-2 and JSM-3 exhibited higher leaf numbers than other varieties, with 4.9 and 5.2 leaves per plant, respectively. As the leaves aged to the milking stage, the average number decreased by 6.78%. In 2025, varieties JSM-2 and JSM-3 exhibited the highest number of leaves, with 5.37 and 5.0 leaves per plant, respectively, which was an increase of 10.95% compared to the heading stage.

The results of the forage yield for the seven triticale varieties at different growth stages showed that the fresh biomass yield initially increased and then decreased as the growth stages progressed. Each variety reached its maximum yield at the flowering stage, followed by a decrease at the milk stage (Figure 1E). These findings were consistent with the growth trends observed in 2025 (Figure 2E). In 2024, the variety JSM-3 exhibited significantly higher fresh biomass yield than the other varieties at all growth stages (*p* < 0.05), with an average range of 14.21~20.72 t·hm^−2^, while QSM-3 had the lowest fresh biomass yield, with only 5.96 t·hm^−2^ at the booting stage, significantly lower than the other varieties (*p* < 0.05). In 2025, the fresh biomass yield of QSM-8 reached its peak, at 20.04 t·hm^−2^. This was significantly higher than that of other varieties (*p* < 0.05). QSM-3 still maintained the lowest fresh biomass yield, at a minimum of 5.64 t·hm^−2^. The dry matter yield of the tested varieties showed a continuous increasing trend, reaching its peak at the milk stage, which was 56.50% higher than at the booting stage. The average dry matter yield reached 6.59 t·hm^−2^. There was no significant difference in dry matter yield between QSM-8 and JSM-3 (*p* > 0.05), with yields of 8.83 and 8.75 t·hm^−2^, respectively, (Figure 1F). QSM-3 had the lowest yield at all growth stages, significantly lower than the other varieties (*p* < 0.05). In 2025, the average dry matter yield reached 6.38 t·hm^−2^, which was slightly lower than that in 2024 (Figure 2F). The dry matter content of each variety exhibited a gradual increase across the four growth periods in 2024 and 2025. Notably, the average dry matter content during the milk stage increased by 17.67% compared to the booting stage in 2024, and by 34.27% in 2025. Among the varieties studied, QSM-8 demonstrated a significantly higher average dry matter content than the others (Figure 1G). The varieties with the lowest dry matter content during the booting and heading stages were JSM-2 and QSM-3, with 27.84% and 31.72%, respectively. During the flowering stage, there was no significant difference between QSM-8 and JSM-3 (*p* > 0.05). During the milk stage, JSM-3 ranked second with a content of 36.62%. The value of JSM-3 reached its peak during the flowering stage, at 34.58% (Figure 2G). Meanwhile, the value of QSM-2 reached its peak during the milk stage, at 45.54%. There were no significant differences among the various varieties during the milk stage.

### 2.3. Effects of Different Varieties and Growth Stages on the Nutritional Quality of Triticale

The two-way ANOVA analysis (Table 3) revealed that the variety significantly affected the nutritional quality of triticale in 2024 (*p* < 0.05). The growth stage had no significant effect on ether extract but showed a significant impact on other nutritional indicators (*p* < 0.05). The interaction between variety and growth stage had a significant effect on soluble sugar, neutral (acid) detergent fiber, relative feed value, and relative forage quality (*p* < 0.05), a significant effect on crude protein, and no significant effect on crude ash and ether extract in 2024 (*p* > 0.05). The two-way ANOVA analysis (Table 4) revealed that both variety and growth stage had a significant impact on the nutritional quality in 2025, except for ether extract. The interaction between variety and growth stage had a significant effect on crude protein, soluble sugar, neutral detergent fiber, acid detergent fiber, relative feeding value, relative forage quality (*p* < 0.05), but had no significant effect on crude ash and ether extract in 2025 (*p* > 0.05).

According to the analysis of CA content and CP content across various growth stages (Table 5), the average CA content ranged from 4.74% to 7.85% in 2024. The CA content during the booting stage was the highest compared to the other three stages, with JSM-3 reaching 9.13%, significantly higher than other varieties (*p* < 0.05). Varieties QSM-7 and QSM-8 showed the highest CA content during the flowering and milking stages, with values of 6.67% and 5.83%, respectively, in 2024. In 2025, the average CA content ranged from 5.03% to 7.25%. The content during the ripening period was slightly higher than that observed in 2024. The CA content of QSM-1 was the lowest, at only 3.87%. According to the analysis of CP content across different growth stages, the average CP content ranged from 5.98% to 12.64%, showing a decreasing trend as the growth stages progressed. QSM-7 exhibited the highest CP content during the booting stage, with no significant difference compared to QSM-8, while QSM-2 had the lowest content at only 10.14% in 2024. During the heading stage, QSM-8 had the highest average content for that period, significantly higher than QSM-1. At the flowering stage, the CP content decreased to 7.70%, with no significant differences among varieties (*p* > 0.05). By the milk stage, the average CP content further decreased to 5.98%, with JSM-2 showing a relatively higher content, showing no significant differences compared to QSM-7 and JSM-3, while QSM-1 had the lowest CP content at only 4.37%. In 2025, QSM-7 still maintained the highest CP content of 14.52%, which was lower than that in 2024. QSM-1 had the lowest content throughout all the reproductive periods, at only 5.87%.

According to the analysis of EE and SS content across various growth stages (Table 6), the average EE content ranged from 1.27% to 1.36%. QSM-8 consistently maintained higher levels during the milking stage, heading stage, and flowering stage, with values of 1.57%, 1.56%, and 1.44%, respectively, in 2024. There were no significant differences among varieties during the flowering stage. During the milking stage, QSM-2 had the highest EE content, reaching 1.49%, while JSM-2 and JSM-3 had the lowest EE content, significantly lower than other varieties (*p* < 0.05). In 2025, the content of QSM-2 was the highest during the booting stage, reaching 1.70%, which was significantly higher than that of other varieties. QSM-7 had the lowest content during the flowering stage, at only 1.13%. According to the analysis of SS content across various growth stages, the SS content gradually increased as the reproductive period progressed in 2024 and 2025. In 2024, the content of QSM-7 reached its peak at 45.62% during the ripening stage. In 2025, the content of QSM-1 was the highest at 49.22%.

Analysis of NDF and ADF content across different growth stages showed that the NDF content of the tested varieties ranged from 52.47% to 62.26% in 2024 (Table 7). QSM-8 had the lowest NDF content during the booting, heading, and milk stages, with values of 57.00%, 57.73%, and 47.20%, respectively. JSM-3 reached the highest NDF content during the heading stage at 67.07%, significantly higher than that of the other varieties (*p* < 0.05). During the milk stage, JSM-2 had the highest NDF content at 58.17%, showing no significant difference compared to QSM-3 at the same stage. In 2025, the average NDF content ranged from 53.77% to 59.72%. The NDF content of JSM-3 was the highest, at 61.99%, which showed no significant difference from QSM-2. QSM-8 had the lowest content, at only 48.54%. The ADF content averaged between 30.30% and 34.81% across the four periods, with the lowest content observed during the milk stage. There were no significant differences among the varieties (*p* > 0.05). QSM-1 and QSM-8 had the lowest ADF content during this period, while JSM-3 showed the highest value at the heading stage, reaching 37.83%, significantly higher than that of the other varieties (*p* < 0.05). In 2025, the average ADF content ranged from 31.03% to 34.10%. The JSM-3 variety also had the highest content, reaching up to 36.44% at the milk stage, while QSM-3 had the lowest content, merely 29.68%.

The analysis of RFV and RFQ across different varieties and growth stages showed significant differences among the tested forage triticale varieties (Table 8). The average RFV at the booting stage was 99.01, with JSM-2 being the highest, surpassing other varieties during the same period in 2024. The RFV reached its peak at the milk stage, averaging 117.06, which was a 15.42% increase compared to the booting stage. Quality assessments identified QSM-8 as demonstrating significantly superior forage indices compared to QSM-3 and JSM-2 contemporaries (*p* < 0.05), followed by QSM-1 (128.11). No significant differences were observed among QSM-2, QSM-3, and JSM-3 genotypes (*p* > 0.05). In 2025, the average RFV ranged from 98.79 to 113.92. QSM-8 had the highest RFV, reaching 124.13, which was not significantly different from QSM-1 (*p* > 0.05). RFQ values exhibited inter-varietal variation ranging from 73.10 to 82.47, with distinct patterns among different varieties at different periods in 2024. Peak RFQ values were consistently observed during the flowering stage, which is consistent with the research results obtained in 2025. QSM-1 and QSM-8 displayed significantly elevated means of 99.28 and 98.06, respectively, outperforming other cultivars in 2024 (*p* < 0.05). Conversely, QSM-2 manifested the lowest RFQ value (57.76) during milking stage development, significantly lower than that of other varieties (*p* < 0.05). In 2025, the PFQ of QSM-8 and QSM-2 was higher than that of other varieties, at 163.67 and 162.71, respectively, significantly higher than those of other varieties (*p* < 0.05). JSM-3 had the lowest PFQ, at only 108.93.

### 2.4. Comprehensive Evaluation of Triticale Varieties at Different Growth Stages

#### 2.4.1. Pearson Correlation Analysis

The Pearson correlation analysis of the relationship between production performance and nutritional quality of triticale revealed the following results (Figure 3). In 2024, plant height had a significantly positive correlation with stem diameter, and fresh biomass yield (*p* < 0.01). The number of leaves had a significant positive correlation with dry matter yield (*p* < 0.01). The number of tillers was significantly negatively correlated with dry matter content (*p* < 0.05). In 2025, plant height had a significant positive correlation with stem diameter (*p* < 0.01). The number of leaves was significantly positively correlated with dry matter yield. The number of tillers was significantly negatively correlated with dry matter content (*p* < 0.05). Stem diameter was significantly positively correlated with fresh biomass yield (*p* < 0.01).

#### 2.4.2. Principal Component Analysis (PCA)

The principal component analysis results of 15 related indicators for the triticale (Table 9) showed that the cumulative variance contribution of the first and second principal components reached 66.1% and 65.34 in 2024 and 2025, indicating that the first two principal components can effectively explain most of the information in the original variables. On the PC1 dimension, the original variables fresh biomass yield, dry matter yield, plant height, number of leaves, and dry matter content had higher loadings, indicating that these indicators were positively correlated with component 1 in 2024 (Figure 4). Plant height, the number of leaves, dry matter yield, and dry matter content had a significant positive loading, while fresh biomass yield had a significant loading in 2025. The first principal component tends to be related to the growth performance and yield characteristics of forage, meaning that plant height and the production capacity of dry matter yield were the core factors influencing PC1 in 2024. Over the two-year principal component analysis, both dry matter yield and dry matter content had significant positive loadings on PC1, indicating that these two indicators play important roles in crop growth. On the PC2 dimension, the original variables ADF, NDF, stem diameter, and tiller number exhibited positive loadings, while EE and RFV showed a weak negative correlation in both components. The second principal component reflected a greater inclination toward factors related to nutritional quality mainly integrating the variation information of nutritional components, fiber content, and tillering ability. The fresh biomass yield had a significant load on PC2, indicating that there was a negative correlation between the fresh biomass yield and other indicators. In 2024, the number of tillers and stem diameter had a significant positive load on PC1, while in 2025, they had a significant positive load on PC2. In 2024, plant height and leaf number had a significant positive load on PC2, while in 2025, they had a significant positive load on PC1. Based on the principal component analysis conducted in 2024 and 2025, the top three indicators in the comprehensive ranking were SS, dry matter content, and RFV. The MKS stage is concentrated in the positive direction region of PC1, indicating more prominent performance in yield-related traits (such as dry matter yield, etc.) dominated by PC1. As the reproductive period progresses, the FLS stage gradually shifts towards the negative performance direction, while the BTS and HDS stages are concentrated in the negative direction range. Biomass accumulation increases, and performance in production and yield factors (such as fresh biomass yield, number of green leaves, etc.) is excellent.

#### 2.4.3. TOPSIS Model Comprehensive Evaluation

The results of the TOPSIS evaluation method indicate that the comprehensive score indices and rankings of each variety reflect their relative proximity to the positive and negative ideal solutions (Table 10). The research results indicate that the variety QSM-8 achieved the highest comprehensive score index (0.614), ranking first, indicating that its distance to the negative ideal solution is significantly greater than its distance to the positive ideal solution, making it the closest to the positive ideal solution. JSM-3 and QSM-1 ranked second and third, respectively, demonstrating superior performance. The lower ranking of QSM-3 indicates its highest degree of deviation from the optimal solution. The overall ranking results validate the rationale of the TOPSIS model, with the performance differences among samples primarily determined by the relative proportions of distances to the positive and negative ideal solutions. A higher comprehensive score index represents a closer proximity to the ideal solution and greater decision-making value. The harvest of the variety during the flowering period is superior to that at other periods, with the highest ideal closeness coefficient (Ci) and the comprehensive score at the milk maturity stage being second. According to Figure 5, the top five varieties ranked by comprehensive evaluation value are QSM-8 > JSM-3 > QSM-1 > JSM-2 > QSM-7 > QSM-1.

## 3. Discussion

Several national research institutes are focused on enhancing forage productivity and developing new forage cultivars by identifying adaptable, high-quality forage accessions. This is primarily achieved through the introduction and evaluation of various options within their forage development programs [11]. The production performance of forage grass is the result of the combined effect of their genetic characteristics and various external factors. Affected by factors such as climate environment and cultivation measures, the same forage variety may exhibit different growth characteristics in different regions [12].

### 3.1. Influence of Period of Fertility and Variety on the Production Performance of Triticale

Key agronomic traits such as plant height, stem diameter, tiller number, and green leaf count are crucial indicators for evaluating the growth and development of forage and are critical factors in assessing forage production performance [13]. Plant height is of great significance in grassland management and can improve feed efficiency in the livestock production system [14]. Selecting forage varieties with a large number of tillers, a high number of leaves, and moderate stem thickness is of great significance as it can not only rapidly increase aboveground biomass, but also provide high-quality feed for livestock. The results of this study indicate that the plant height of triticale progressively increases with the growth stages. However, the growth rate slows down during the milk stage, with an increase of 24.99% compared to the booting stage, and a total increase of 25% over the booting stage in 2024. The varieties JSM-3 and QSM-8 exhibit significantly greater plant heights than other varieties, which is attributed to their stronger apical dominance and the synchronous elongation of stems and spike differentiation [15,16]. The plant height of JSM-3 was significantly higher than that of other varieties at all four growth stages, reaching its maximum value of 151.20 cm during the milk stage. The increase in plant height for varieties QSM-1, QSM-2, and QSM-3 slowed down during the milk stage. This may be related to its earlier entry into the reproductive growth phase [17,18]. The number of tillers of triticale gradually decrease. As the growth period progresses, the plant height increases, and the water content of the stems and leaves decreases. Ineffective tillers gradually die out and decrease. The degree of lignification of the stems increases, and the internodes at the base of the stems gradually become full. The thickness of the stem decreases. The number of green leaves is the highest during the flowering period and then decreases. The leaves gradually age, and the lower leaves of the plant fall off [19]. Among forage grasses, varieties with larger stem thicknesses may have a higher degree of plant lignification, often accompanied by an increase in lignin content. When lignin combines with fibers, it will reduce the digestibility of forage grasses. Although it affects palatability, the increase in stem biomass can enhance the overall forage yield [20,21]. Forage biomass yield is a major target in forage crop improvement [22].

Forage yield is a key indicator for evaluating the production performance of forage, and its level directly affects economic benefits. As a hybrid grass of the Poaceae family, triticale exhibits strong stress resistance and tolerance to poor soil conditions, enabling it to grow normally in regions with harsh natural environments and achieve high forage yield. In this study, the fresh biomass yield of triticale significantly increased with the advancement of the growth period, reaching the maximum value during the flowering period in 2024 and 2025. The variety JSM-3 and QSM-8 had the highest grass yield during the flowering period. During the milk maturity period, due to the increased stem lignification causing leaf shedding, the fresh biomass yield slightly decreased. This trend is consistent with the studies of Zhao et al. and Sun et al. [23,24]. In this study, the dry matter yield continued to rise, peaking at the milk stage. This aligns with the findings of Zhu et al. [25], which indicate that under the ecological conditions of Jiangsu, the fresh and hay yields of different types of triticale gradually increase as the growth period progresses. The fresh biomass yield reaches its maximum at the flowering stage, with a few varieties peaking at the booting or grain formation stages, while the dry weight yield reaches its maximum at the filling or maturity stages, with variations among varieties and types. Therefore, from the flowering stage to the milk stage, triticale grass yields are substantial, making it suitable for the production of hay and silage.

### 3.2. The Influence of Growth Period and Breed on the Nutritional Quality of Triticale

Nutritional quality is a crucial factor in evaluating the forage value of a pasture. The content of ADF and NDF are key indicators for assessing the palatability and digestibility of forage [26]. A lower content of these components indicates improved palatability and higher digestibility, playing an indispensable role in the feed value. The content of CP is a core indicator for evaluating the nutritional quality of forage [27]. Under natural growth conditions, the increase in dry matter yield of forage is often accompanied by a decrease in protein content and an increase in NDF, especially when the increase in yield stems from the accumulation of stem biomass or the advancement of growth stages [28,29]. Increasing the crude protein content and reducing the crude fiber content are important approaches to enhance the nutritional value and improve the nutritional quality of pasture and forage crops [30,31]. This study indicates that the CP content of triticale significantly decreases with the advancement of the growth period in 2024 and 2025. The content reaches its peak (10.63–15.09%) during the booting stage, when the stems and leaves grow rapidly and nitrogen accumulation is significant. It drops to 5.49–9.26% during the flowering stage and further decreases to 4.37–9.35% during the milk maturity stage in 2024. It drops to 5.87–8.61% during the flowering stage and further decreases to 4.97–7.31% during the milk maturity stage in 2025. The CP content of QSM-8 was 14.89% at the booting stage and decreased to 5.55% at the milk maturity stage. The CP content of QSM-7 reached 15.09% at the booting stage, which was consistent with the trend of the study by Liang Sun et al. [24] (CP 1.62–7.49%), but the overall value was relatively low, which might be related to the limited nitrogen absorption in the highly cold mountainous area environment of Guizhou. However, it was significantly higher than in the highly cold pastoral areas of Gansu Province (7.40–13.31%), indicating a strong nitrogen enrichment capacity during the booting stage and is suitable for promotion as a high-protein forage variety. Crude ash content is an important indicator for evaluating the quality of forage. Excessive crude ash content indicates poor forage quality [32]. In this study, the contents of CA, NDF, and ADF gradually decreased and were the lowest at the milk maturity stage. Among them, the contents of QSM-8 and QSM-1 were the lowest, and there was no significant difference (*p* < 0.05). Harvesting of triticale during the flowering stage could achieve a balance between dry matter accumulation and crude protein content. How to convert the experimental results into actual animal nutrition in the future and whether there are additional feed quality indicators for assessment will remain to be verified. This will enhance the comprehensiveness and accuracy of nutritional evaluation.

### 3.3. Comprehensive Evaluation of the Triticale Varieties

The TOPSIS model, introduced by Wang and Yoon in 1981, is a robust method for comparing and selecting multiple indicators and schemes, effectively addressing multi-objective decision-making problems with a limited set of alternatives [33]. This study conducted a comprehensive evaluation of the four growth stages of seven feed triplet rye varieties. The fresh biomass yield of JSM-3 at the flowering stage was the highest, and the hay yield of QSM-8 at the milk maturity stage was the highest. Among them, the contents of CP, CA, NDF, and ADF were relatively low. QSM-8 had the best comprehensive performance and the highest ideal closeness, followed by variety JSM-3. Forage yield is an important indicator reflecting the production performance of forage, and nutritional value is the primary determinant of forage quality. In this study, in the Qaidam area, if the goal is to obtain high-protein forage, varieties with high scores should be prioritized and harvested before the flowering period.

## 4. Materials and Methods

### 4.1. General Conditions of Experimental Site

The test site is located at the National Grass Variety Regional Test Station in Gahai Town, Delhi City, Qinghai Province (Figure 6). This station is situated in the northern part of the Haixi Mongolian and Tibetan Autonomous Prefecture, on the northeastern edge of the Qaidam Basin, at a latitude of 37°22′ N and a longitude of 97°15′ E, with an altitude of 2842.3 m. It features a plateau continental climate characterized by thin air, strong solar radiation, and significant temperature fluctuations between morning and evening. The annual average accumulated temperature of ≥0 °C totals 2363.90 °C, with an average of 210 days. The accumulated temperature of ≥10 °C is 660.0 °C an average of 113 days. The area receives 3182.80 h of sunshine annually, with an annual radiation amount of 693.33 kJ·cm^−2^. The annual average precipitation is 181.80 mm, while the annual evaporation is 2370 mm [1]. The frost-free period lasts between 90 and 110 days, with approximately 44.1 days of strong winds annually and an average wind speed of 3.0 m·s^−1^. The meteorological information of the triticale variety planting season in 2024 and 2025 is shown in Figure 7. The soil type is classified as salinized cultivated chestnut soil, with the plow layer containing a salt content exceeding 10.0 g/kg; in some areas, the salt content can reach up to 52.7 g/kg, accompanied by a salt crust of about 2 cm. The pH level ranges from 8.6 to 8.9, with chloride being the predominant salt composition. The basic soil nutrients are shown in Table 11 [34].

### 4.2. Plant Materials

This experiment involves a total of seven triticale varieties. This trial selected seven forage triticale varieties as experimental materials, namely QSM-1 (Qingsimai No. 1), QSM-2 (Qingsimai No. 2), QSM-3 (Qingsimai No. 3), QSM-7 (Qingsimai No. 7), QSM-8 (Qingsimai No. 8), JSM-2 (Jisimai No. 2), and JSM-3 (Jisimai No. 3). These cultivars, provided by Qinghai University and Ningxia Xibei Agroforestry and Animal Husbandry Technology Co., Ltd. (7–9 Northwest Agricultural Materials Market, Qing he North Street, Yinchuan, Ningxia, China), are regionally adapted to spring-sown conditions. Prior to experimentation, all seeds underwent quality validation, confirming compliance with national seed certification standards through verified purity (>98%), germination rate (>92%), and vigor index (>8.0) metrics. The detailed information on these varieties can be found in Table 12.

### 4.3. Experimental Design

A randomized complete block design (RCBD) with three replications was implemented to evaluate seven triticale varieties. Each experimental unit consisted of 3 m × 5 m plots (15 m^2^) separated by 50 cm buffer zones to minimize edge effects. The entire trial was surrounded by two border rows of triticale cultivar. Seedbed preparation included moldboard plowing to a depth of 15 cm followed by two passes with a disk harrow. Basal fertilization was applied before planting at rates of 90 g·m^−2^ diammonium phosphate and 150 g·m^−2^ urea, incorporated into the upper 15 cm soil layer. The experiment was conducted over two years in 2024 and 2025. Row sowing was used, with a sowing depth of 4–5 cm and a row spacing of 20 cm. A sowing rate of 200 kg per hectare was employed, consistent with the recommended density for optimal plant population and yield of triticale in the region [35,36]. After that, germination management included three mechanical cultivations (15, 30, and 45 days after sowing) for weed control and soil aeration. Growth stages were monitored biweekly using the BBCH scale. When more than 80% of the plants in each plot reached the target growth stage, a 1-square-meter quadrat was harvested from the center of the plot using a forage sampler, with the stubble height maintained at 5 cm.

### 4.4. Determination of Indicators and Methods

Phenological stage: Sowing time of the crop was recorded, and the durations of key growth stages—including the booting stage, heading stage, flowering stage, and milky ripe stage—were observed and documented. The identification criteria for each growth period are as follows: the initial stage is when 10% of the plants enter a certain growth stage, the peak stage is when 50% is reached, and the final stage is when 80% is reached.

Plant height: Ten individual plants were randomly selected and tagged in each plot. The natural height of each plant, measured from ground level to the apical meristem (or highest point), was recorded. The mean plant height per variety was calculated from measurements taken across three replicate plots.

Tillering number: The number of tillers were counted on ten randomly selected representative plants per plot, excluding those in the border rows.

Stem diameter: Stem diameter was measured at each growth stage using a vernier caliper. The width of the thickest part of the stem was recorded.

Number of green leaves: Ten representative plants were randomly selected from the inner area of each plot, and the number of green leaves per plant was recorded.

Grass yield: During the booting, heading, flowering, and milking stages of the crop, all plants within a randomly selected area of 1 m × 1 m in each plot were mowed, weeds were removed, and the above-ground plant material was weighed to calculate the fresh biomass yield per hectare. All fresh weight samples were then taken to the laboratory, air-dried naturally, and then weighed to obtain the dry matter yield.

Nutritional quality: The air-dried samples collected at different growth stages were ground and sieved through a 40-mesh filter to assess their nutritional quality. Each sample was analyzed in triplicate. The crude ash (CA) content was determined using the high-temperature ash method, while the crude protein (CP) content was assessed through the Kjeldahl nitrogen determination method. The ether extract (EE) content was quantified via the Soxhlet fat extraction method. Additionally, the acid detergent fiber (ADF) and neutral detergent fiber (NDF) contents were measured using the filter bag method. The content of soluble sugar (SS) was determined using the anthrone colorimetric method. The relative feeding value (RFV) and relative forage quality (RFQ) were calculated using the following formulas [37]:(1)$$\mathrm{Relative}\;\mathrm{feed}\;\mathrm{value}:\;\mathrm{RFV}=\frac{\left(\frac{120}{\mathrm{NDF}}\times \left(88.9-0.779\;\times \mathrm{ADF}\right)\right)}{1.29}$$(2)$$\mathrm{Relative}\;\mathrm{forage}\;\mathrm{quality}:\;\mathrm{RFQ}=\frac{\left(\mathrm{DMI},\;\%\;\mathrm{of}\;\mathrm{BW}\right)\times \left(\mathrm{TDN},\;\%\;\mathrm{of}\;\mathrm{DM}\right)}{1.23}$$(3)$$\mathrm{Dry}\;\mathrm{matter}\;\mathrm{intake}:\;\mathrm{DMI}\left(\%,\;\mathrm{of}\;\mathrm{BW}\right)=\frac{120}{\mathrm{NDF},\;\%\;\mathrm{of}\;\mathrm{DM}}$$(4)Total digestible nutrient: TDN=88.9−(0.779×NDF,% of DM)

### 4.5. Data Statistics and Analysis

The experimental data were organized and analyzed using Excel 2021 software. The normality and homogeneity of variance of the data were assessed with SPSS 23.0 (SPSS for Windows, Chicago, IL, USA). A one-way analysis of variance (ANOVA) and interaction analysis were performed on different growth stages of seven varieties, with significant differences analyzed using Duncan’s multiple comparison test. The production performance and nutritional quality of the various varieties were evaluated for significance at the 0.05 level. Principal component graphs were generated using Origin 2023 (OriginLab Corporation, Northampton, MA, USA), with major indicators determined at a 95% confidence interval. The geographic map was generated using ArcGIS 2022 (Esri Inc., Redlands, CA, USA). software. Results were visualized using cartographic functions to output thematic maps with symbolized classifications and layout elements. The Pearson correlation method in R 4.3.2 was used to examine the relationship between the production performance and nutritional quality of triticale. The indicator data were standardized using the entropy weight coefficient method in Excel 2021, calculating the proportion of main indicator values and processing the information entropy of the indicators. Weights were assigned to the primary evaluation indicators of each variety. The multi-decision TOPSIS model was used to comprehensively analyze the positive and negative ideal solutions for different varieties and growth periods in terms of production performance and nutritional quality. Formula calculations were carried out using Excel 2021 software. The Euclidean distances from each variety to the positive and negative ideal solutions were computed, ultimately determining the Closeness Coefficient (to the Ideal Solution). Within the range of 0 ≤ *Ci* ≤ 1, a larger Ci indicates that the evaluation object is closer to the positive ideal solution, signifying a higher closeness degree and better comprehensive performance. The Euclidean distances from each evaluation object to the positive and negative ideal solutions were calculated as follows:(5)$$\mathrm{Euclidean}\;\mathrm{Distance}\;\mathrm{to}\;\mathrm{Positive}\;\mathrm{Ideal}\;\mathrm{Solution}:\;D_{i}^{+}=\sqrt{\sum_{j=1}^{m}{\left(x_{ij}-x_{j}^{+}\right)}^{2}}$$(6)$$\mathrm{Euclidean}\;\mathrm{Distance}\;\mathrm{to}\;\mathrm{Negative}\;\mathrm{Ideal}\;\mathrm{Solution}:\;D_{i}^{-}=\sqrt{\sum_{j=1}^{m}{\left(x_{ij}-x_{j}^{-}\right)}^{2}}$$(7)$$\mathrm{Closeness}\;\mathrm{Coefficient}:\;C_{i}=\frac{D_{i}^{-}}{D_{i}^{+}+D_{i}^{-}}$$

Among them, *x_ij_*: weighted normalized value of the *i*-th solution on the *j*-th criterion. *x_j_*^+^: optimal value (maximum for benefit criteria, minimum for cost criteria after positive normalization) of the *j*-th criterion in the positive ideal solution. *x_j_*^−^: worst-case value (minimum for benefit criteria, maximum for cost criteria after positive normalization) of the *j*-th criterion in the negative ideal solution. *Ci*: it quantifies the relative proximity of the *i*-th solution to the positive ideal solution.

Solutions are ranked in descending order of *Ci*, with higher values indicating better proximity to the ideal solution.

## 5. Conclusions

This study performed a comprehensive analysis of 15 indicators such as production performance and nutritional quality, across seven triticale varieties at different growth stages. The results indicate that the QSM-8 variety is well-suited for promotion and cultivation in the Qaidam region, with an optimal harvest window spanning from the flowering period to the milk maturity period. In conclusion, while this study provides valuable insights into the introduction and cultivation of varieties, it is important to acknowledge its limitations. The data were collected from a single location, which may restrict the generalizability of the results. Given the significant climate variability in the region, future research will be considered to confirm these findings in subsequent studies across multiple locations. This research can provide both theoretical demonstration and practical reference for forage planting and breeding in cold and arid areas globally.

## Figures and Tables

**Figure 1 plants-14-02942-f001:**
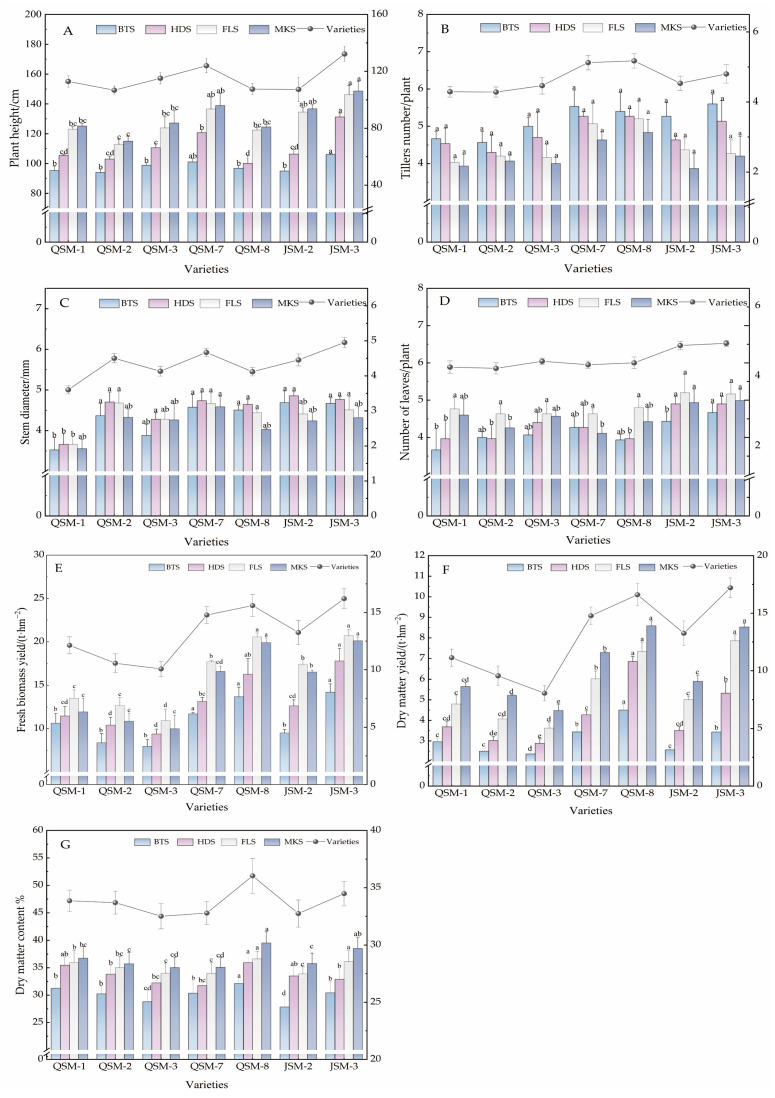
Agronomic traits and grass yield of the participating triticale varieties at different growth periods in 2024. (**A**) Plant height of the triticale variety; (**B**) tillering number of the triticale variety; (**C**) stem thickness of the triticale variety; (**D**) number of green leaves of the triticale variety; (**E**) fresh biomass yield of the triticale variety; (**F**) dry matter yield of the triticale variety; (**G**) dry matter content of the triticale variety. Note: Different lowercase letters indicate significant differences among different varieties within the same growth period at the 0.05 level.

**Figure 2 plants-14-02942-f002:**
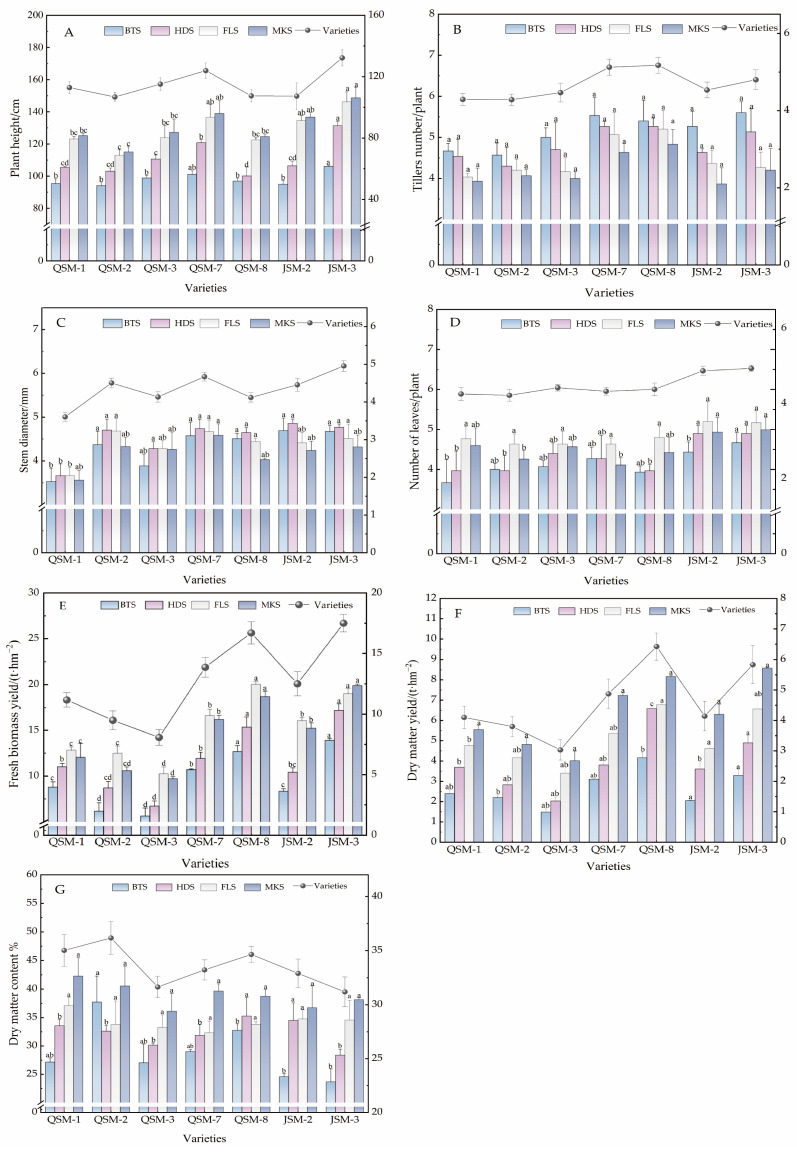
Agronomic traits and grass yield of the participating triticale varieties at different growth periods in 2025. (**A**) Plant height of the triticale variety; (**B**) tillering number of the triticale variety; (**C**) stem thickness of the triticale variety; (**D**) number of green leaves of the triticale variety; (**E**) fresh biomass yield of the triticale variety; (**F**) dry matter yield of the triticale variety; (**G**) dry matter content of the triticale variety. Note: Different lowercase letters indicate significant differences among different varieties within the same growth period at the 0.05 level.

**Figure 3 plants-14-02942-f003:**
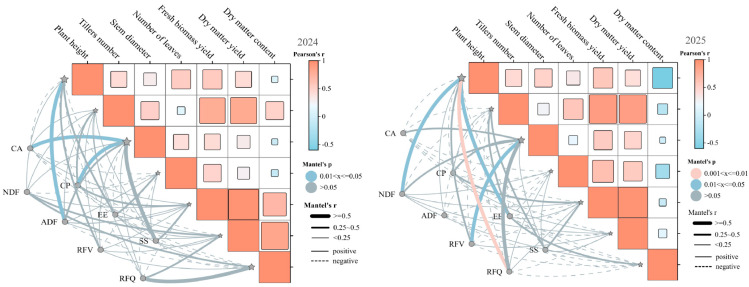
Mantel test correlation analysis of production performance and nutritional quality of different triticale varieties in 2024 and 2025.

**Figure 4 plants-14-02942-f004:**
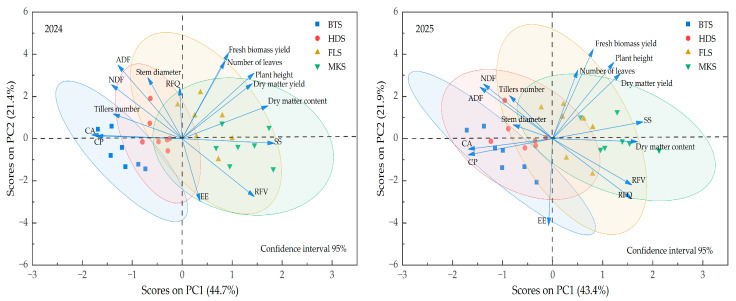
Analysis of the main composition of different indicators of triticale varieties in 2024 and 2025. The area enclosed by the blue ellipse represents the booting stage (BTS), the area enclosed by the red ellipse represents the heading stage (HDS), the area enclosed by the yellow ellipse represents the flowering stage (FLS), and the green area represents the milk-ripe stage (MKS).

**Figure 5 plants-14-02942-f005:**
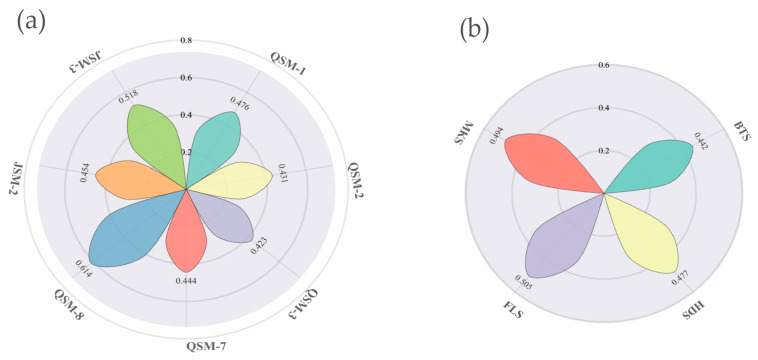
Analysis of the main composition of different indicators of triticale varieties. (**a**) represents the comprehensive score index of different triticale varieties; (**b**) represents the comprehensive score index at different growth stages.

**Figure 6 plants-14-02942-f006:**
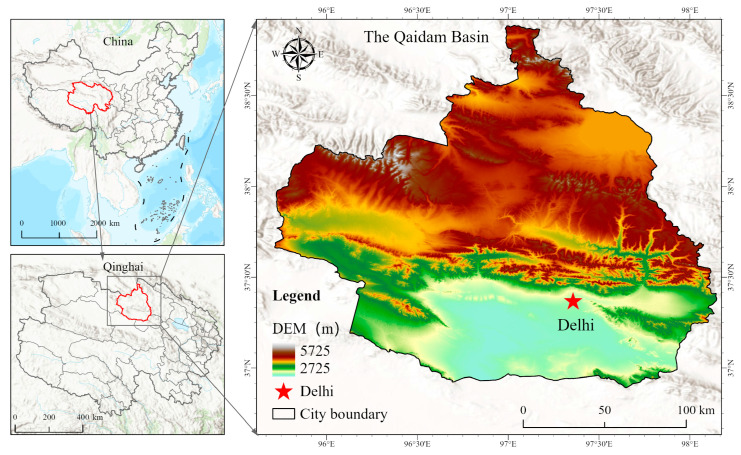
Geographic location of the *experimental site*.

**Figure 7 plants-14-02942-f007:**
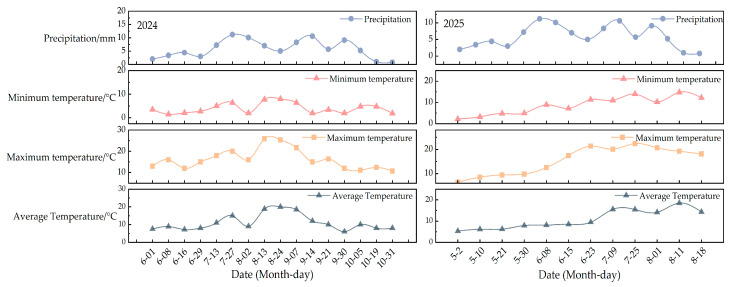
Meteorological data during the cropping seasons in 2024 and 2025.

**Table 1 plants-14-02942-t001:** Variance analysis of production performance of triticale across varieties and growth periods in 2024.

Item	Factor	Sum of Squares	Degree of Freedom	Mean Square	F	*p*
Plant height	Variety	6760.33	6.00	1126.72	28.09	0.00
	Growth periods	18,527.92	3.00	6175.97	154.00	0.00
	Variety × Growth periods	3443.49	18.00	191.31	4.77	0.00
Tiller number	Variety	9.98	6.00	1.66	3.45	0.01
	Growth periods	10.45	3.00	3.48	7.23	0.00
	Variety × Growth periods	2.15	18.00	0.12	0.25	1.00
Stem diameter	Variety	13.90	6.00	2.32	11.41	0.00
	Growth periods	0.40	3.00	0.13	0.65	0.59
	Variety × Growth periods	5.17	18.00	0.29	1.42	0.16
Number of leaves	Variety	6.21	6.00	1.04	7.21	0.00
	Growth periods	6.32	3.00	2.11	14.66	0.00
	Variety × Growth periods	2.26	18.00	0.13	0.88	0.61
Fresh biomass yield	Variety	1088.12	6.00	181.35	68.71	0.00
	Growth periods	503.47	3.00	167.82	63.58	0.00
	Variety × Growth periods	47.65	18.00	2.65	1.00	0.47
Dry matter yield	Variety	154.95	6.00	25.82	129.16	0.00
	Growth periods	166.11	3.00	55.37	276.93	0.00
	Variety × Growth periods	12.75	18.00	0.71	3.54	0.00
Dry matter content	Variety	550.98	6.00	91.83	2.40	0.04
	Growth periods	2849.56	3.00	949.86	24.84	0.00
	Variety × Growth periods	786.81	18.00	43.71	1.14	0.34

Note: *p* > 0.05 means not significant; *p* < 0.05 means significant.

**Table 2 plants-14-02942-t002:** Variance analysis of the production performance of triticale across varieties and growth periods in 2025.

Item	Factor	Sum of Squares	Degree of Freedom	Mean Square	F	*p*
Plant height	Variety	4848.26	6.00	808.04	14.84	0.00
	Growth periods	13,299.67	3.00	4433.22	81.41	0.00
	Variety × Growth periods	2721.66	18.00	151.20	2.78	0.00
Tiller number	Variety	9.58	6.00	1.60	1.48	0.20
	Growth periods	3.86	3.00	1.29	1.19	0.32
	Variety × Growth periods	22.01	18.00	1.22	1.129	0.35
Stem diameter	Variety	9.77	6.00	1.63	3.33	0.00
	Growth periods	3.29	3.00	1.10	2.24	0.09
	Variety × Growth periods	8.36	18.00	0.47	0.95	0.53
Number of leaves	Variety	11.63	6.00	1.94	5.14	0.00
	Growth periods	3.25	3.00	1.08	2.87	0.04
	Variety × Growth periods	7.00	18.00	0.39	1.03	0.44
Fresh biomass yield	Variety	892.27	6.00	148.71	122.18	0.00
	Growth periods	468.68	3.00	156.23	128.36	0.00
	Variety × Growth periods	39.18	18.00	2.18	1.79	0.05
Dry matter yield	Variety	120.14	6.00	20.02	99.08	0.00
	Growth periods	159.18	3.00	53.06	262.55	0.00
	Variety × Growth periods	13.26	18.00	0.74	3.64	0.00
Dry matter content	Variety	443.27	6.00	73.88	3.075	0.01
	Growth periods	2554.24	3.00	851.41	35.44	0.00
	Variety × Growth periods	601.99	18.00	33.44	1.39	0.17

**Table 3 plants-14-02942-t003:** Variance analysis of oat nutritional quality of triticale across varieties and growth periods in 2024.

Item	Factor	Sum of Squares	Degree of Freedom	Mean Square	F	*p*
Crude ash	Variety	18.38	6.00	3.06	4.55	0.00
	Growth periods	113.71	3.00	37.90	56.35	0.00
	Variety × Growth periods	14.82	18.00	0.82	1.22	0.28
Crude protein	Variety	79.06	6.00	13.18	6.65	0.00
	Growth periods	502.09	3.00	167.36	84.43	0.00
	Variety × Growth periods	65.29	18.00	3.63	1.83	0.04
Ether extract	Variety	1.03	6.00	0.17	5.67	0.00
	Growth periods	0.12	3.00	0.04	1.38	0.26
	Variety × Growth periods	0.52	18.00	0.03	0.95	0.52
Soluble sugar	Variety	1195.85	6.00	199.31	9.41	0.00
	Growth periods	13,693.24	3.00	4564.41	215.41	0.00
	Variety × Growth periods	1777.14	18.00	98.73	4.66	0.00
Neutral detergent fiber	Variety	284.80	6.00	47.47	4.29	0.00
	Growth periods	1054.76	3.00	351.59	31.74	0.00
	Variety × Growth periods	595.83	18.00	33.10	2.99	0.00
Acid detergent fiber	Variety	100.81	6.00	16.80	3.09	0.01
	Growth periods	229.27	3.00	76.42	14.07	0.00
	Variety × Growth periods	177.52	18.00	9.86	1.82	0.00
Relative feeding value	Variety	1794.33	6.00	299.06	3.70	0.00
	Growth periods	6829.97	3.00	2276.66	28.19	0.00
	Variety × Growth periods	3920.52	18.00	217.81	2.70	0.00
Relative forage quality	Variety	786.52	6.00	131.08	1.98	0.08
	Growth periods	6439.66	3.00	2146.55	32.50	0.00
	Variety × Growth periods	2985.15	18.00	165.84	2.51	0.00

**Table 4 plants-14-02942-t004:** Variance analysis of oat nutritional quality of triticale across varieties and growth periods in 2025.

Item	Factor	Sum of Squares	Degree of Freedom	Mean Square	F	*p*
Crude ash	Variety	61.86	3.00	20.62	43.33	0.00
	Growth periods	18.11	6.00	3.02	6.34	0.00
	Variety × Growth periods	8.18	18.00	0.46	0.96	0.52
Crude protein	Variety	54.57	6.00	9.10	9.56	0.00
	Growth periods	359.91	3.00	119.97	126.04	0.00
	Variety × Growth periods	50.40	18.00	2.80	2.94	0.00
Ether extract	Variety	1.01	6.00	0.17	7.00	0.00
	Growth periods	0.07	3.00	0.02	0.98	0.41
	Variety × Growth periods	0.52	18.00	0.03	1.21	0.29
Soluble sugar	Variety	903.76	6.00	150.63	18.58	0.00
	Growth periods	9084.80	3.00	3028.27	373.54	0.00
	Variety × Growth periods	939.94	18.00	52.22	6.44	0.00
Neutral detergent fiber	Variety	218.93	6.00	36.49	5.20	0.00
	Growth periods	385.24	3.00	128.41	18.31	0.00
	Variety × Growth periods	359.92	18.00	20.00	2.85	0.00
Acid detergent fiber	Variety	58.33	6.00	9.72	3.69	0.00
	Growth periods	111.10	3.00	37.03	14.06	0.00
	Variety × Growth periods	135.35	18.00	7.52	2.86	0.00
Relative feeding value	Variety	1013.86	6.00	168.98	3.97	0.00
	Growth periods	2510.19	3.00	836.73	19.66	0.00
	Variety × Growth periods	1986.33	18.00	110.35	2.59	0.00
Relative forage quality	Variety	3528.66	6.00	588.11	4.17	0.00
	Growth periods	7626.90	3.00	2542.30	18.00	0.00
	Variety × Growth periods	5542.23	18.00	307.90	2.18	0.01

**Table 5 plants-14-02942-t005:** The crude ash and crude protein content of different triticale varieties in 2024 and 2025 at different growth stages.

Variety	Growth Periods	Record of 2024		Record of 2025	
		Crude Ash %	Crude Protein %	Crude Ash %	Crude Protein %
QSM-1	BTS	7.00 ± 0.75 Ca	10.63 ± 1.03 Ca	6.04 ± 0.15 Ca	10.09 ± 0.20 Ca
	HDS	5.53 ± 0.15 Ab	6.81 ± 0.47 Cb	5.53 ± 0.145 Ba	7.18 ± 0.29 Cb
	FLS	4.43 ± 0.23 Ac	5.49 ± 0.28 Ac	4.44 ± 0.29 Bb	5.87 ± 0.23 Bc
	MKS	3.37 ± 0.78 Bd	4.37 ± 0.10 Bd	3.87 ± 0.21 Bb	4.97 ± 0.20 Bd
QSM-2	BTS	7.37 ± 0.97 BCa	10.14 ± 2.48 Ca	7.37 ± 0.57 ABa	9.52 ± 1.13 Ca
	HDS	6.47 ± 0.93 Aab	8.87 ± 0.79 ABab	6.47 ± 0.56 ABab	9.07 ± 0.28 ABa
	FLS	5.17 ± 0.86 Ab	8.16 ± 0.34 Aab	5.25 ± 0.59 ABc	7.94 ± 0.21 ABab
	MKS	4.97 ± 0.35 ABb	6.58 ± 0.49 Ab	5.23 ± 0.14 ABc	6.57 ± 0.23 Ab
QSM-3	BTS	7.57 ± 0.42 BCa	10.91 ± 0.61 BCa	6.81 ± 0.26 BCa	10.73 ± 0.30 Ca
	HDS	6.13 ± 0.15 Ab	8.06 ± 1.08 BCb	5.98 ± 0.26 ABb	8.67 ± 0.42 ABCb
	FLS	5.33 ± 0.64 Aab	7.68 ± 2.17 Abc	5.45 ± 0.38 ABbc	7.93 ± 0.79 ABb
	MKS	4.73 ± 0.68 ABc	5.42 ± 0.60 ABc	4.97 ± 0.15 ABc	5.87 ± 0.29 ABc
QSM-7	BTS	8.20 ± 0.20 ABa	15.09 ± 0.70 Aa	7.68 ± 0.248 ABa	14.52 ± 0.34 Aa
	HDS	6.43 ± 0.61 Ab	8.08 ± 1.86 BCb	6.54 ± 0.24 ABab	8.11 ± 0.58 BCb
	FLS	6.67 ± 1.33 Ab	7.77 ± 3.01 Ab	6.20 ± 0.55 Ab	7.88 ± 1.07 ABb
	MKS	4.87 ± 0.51 ABc	6.59 ± 1.44 Ab	4.87 ± 0.29 ABc	6.72 ± 0.78 Ab
QSM-8	BTS	8.23 ± 0.23 ABa	14.89 ± 0.39 Aa	7.77 ± 0.14 ABa	13.55 ± 0.44 ABa
	HDS	6.53 ± 0.61 Aab	10.84 ± 1.67 Ab	6.64 ± 0.14 ABab	10.36 ± 0.70 Ab
	FLS	5.07 ± 0.15 Ab	7.75 ± 0.90 Ac	5.25 ± 0.28 ABc	7.97 ± 0.32 ABc
	MKS	5.83 ± 2.50 Aab	5.55 ± 0.26 ABd	5.73 ± 0.99 Ac	5.84 ± 0.10 ABd
JSM-2	BTS	7.43 ± 0.25 BCa	13.86 ± 0.92 Aa	7.04 ± 0.25 ABCa	12.50 ± 0.41 Ba
	HDS	7.23 ± 1.86 Aa	8.18 ± 1.93 BCb	7.24 ± 0.25 Aa	7.99 ± 0.82 BCb
	FLS	5.67 ± 0.38 ABab	7.77 ± 3.13 Ab	5.96 ± 0.22 Aab	7.92 ± 1.18 ABb
	MKS	5.13 ± 0.68 ABb	6.75 ± 1.34 Ab	5.36 ± 0.25 Ab	6.56 ± 0.84 Ab
JSM-3	BTS	9.13 ± 0.21 Aa	12.92 ± 0.61 ABa	8.08 ± 0.54 Aa	12.42 ± 0.31 Ba
	HDS	6.43 ± 0.51 Ab	9.56 ± 1.10 ABb	5.88 ± 0.54 ABa	9.37 ± 0.39 ABb
	FLS	5.77 ± 0.49 ABb	9.26 ± 1.85 Ab	5.41 ± 0.35 ABab	8.61 ± 0.09 Ab
	MKS	4.30 ± 0.10 ABc	6.59 ± 0.58 Ac	5.20 ± 0.23 ABb	7.31 ± 0.28 Ab

Note: Different uppercase letters indicate significant differences among different triticale varieties within the same growth stage (*p* < 0.05); different lowercase letters indicate significant differences for the same triticale variety across different growth stages (*p* < 0.05).

**Table 6 plants-14-02942-t006:** The ether extract and soluble sugar content of different triticale varieties in 2024 and 2025 at different growth stages.

Variety	Growth Periods	Record of 2024		Record of 2025	
		Ether Extract %	Soluble Sugar %	Ether Extract %	Soluble Sugar %
QSM-1	BTS	1.25 ± 0.04 ABa	11.61 ± 3.54 ABd	1.54 ± 0.17 ABa	14.40 ± 0.59 ABd
	HDS	1.44 ± 0.20 ABa	21.00 ± 2.74 Ac	1.48 ± 0.13 ABa	19.73 ± 0.71 Bc
	FLS	1.26 ± 0.06 Aa	42.19 ± 1.57 Ab	1.41 ± 0.02 BCa	41.01 ± 0.50 Ab
	MKS	1.43 ± 0.20 ABa	52.41 ± 5.51 Aa	1.38 ± 0.03 Ba	49.22 ± 0.50 Aa
QSM-2	BTS	1.41 ± 0.27 ABa	13.55 ± 6.28 Ab	1.7 ± 0.14 Aa	16.61 ± 0.24 Ac
	HDS	1.44 ± 0.21 ABa	18.04 ± 7.6 Ab	1.64 ± 0.05 Aa	16.06 ± 1.97 Bc
	FLS	1.41 ± 0.28 Aa	33.70 ± 3.76 Ba	1.73 ± 0.12 Aa	33.70 ± 2.17 BCb
	MKS	1.49 ± 0.02 Aa	42.98 ± 3.14 ABa	1.53 ± 0.01 Aa	40.96 ± 1.08 Ba
QSM-3	BTS	1.41 ± 0.10 ABa	6.64 ± 4.34 ABCb	1.57 ± 0.09 ABab	9.82 ± 0.94 Cc
	HDS	1.24 ± 0.08 Ba	26.96 ± 2.81 Aa	1.42 ± 0.05 ABb	26.71 ± 1.49 Ab
	FLS	1.40 ± 0.15 Aa	36.22 ± 7.18 Ba	1.67 ± 0.08 ABa	34.96 ± 2.79 Ba
	MKS	1.29 ± 0.16 ABa	37.21 ± 7.12 Ba	1.44 ± 0.05 ABab	36.11 ± 3.38 BCa
QSM-7	BTS	1.03 ± 0.13 Bc	2.24 ± 1.23 Cd	1.36 ± 0.16 ABa	12.09 ± 0.78 BCc
	HDS	1.32 ± 0.10 ABab	17.00 ± 5.58 Ac	1.42 ± 0.09 ABa	16.08 ± 2.42 Bc
	FLS	1.13 ± 0.07 Abc	28.53 ± 4.92 Bb	1.13 ± 0.04 Ca	29.56 ± 0.77 CDb
	MKS	1.42 ± 0.11 ABa	45.62 ± 0.66 ABa	1.45 ± 0.055 ABa	40.17 ± 0.03 Ba
QSM-8	BTS	1.57 ± 0.08 Aa	5.03 ± 3.32 BCd	1.57 ± 0.05 ABa	9.25 ± 1.19 Cd
	HDS	1.56 ± 0.13 Aa	20.09 ± 3.00 Ac	1.56 ± 0.08 ABa	19.16 ± 1.23 Bc
	FLS	1.44 ± 0.09 Aa	37.29 ± 2.78 Ba	1.42 ± 0.10 BCa	32.48 ± 1.25 BCa
	MKS	1.37 ± 0.14 ABa	25.32 ± 0.21 Cb	1.46 ± 0.06 ABa	25.24 ± 0.09 Db
JSM-2	BTS	1.14 ± 0.46 Ba	6.13 ± 6.22 ABCc	1.25 ± 0.17 Ba	9.24 ± 2.14 Cd
	HDS	1.35 ± 0.12 ABa	16.14 ± 6.33 Abc	1.46 ± 0.04 ABa	15.03 ± 2.54 Bc
	FLS	1.14 ± 0.33 Aa	21.70 ± 3.38 Ab	1.25 ± 0.167 Ca	26.03 ± 0.29 Db
	MKS	1.20 ± 0.16 Ba	36.30 ± 4.80 Ba	1.36 ± 0.01 Ba	32.58 ± 1.16 Ca
JSM-3	BTS	1.17 ± 0.07 ABa	2.28 ± 1.65 Cc	1.42 ± 0.01 ABa	10.49 ± 1.149 Cd
	HDS	1.18 ± 0.16 ABa	17.23 ± 6.13 Ab	1.32 ± 0.04 Ba	17.47 ± 3.64 Bc
	FLS	1.12 ± 0.08 Aa	23.30 ± 6.62 Ab	1.33 ± 0.05 Ca	29.90 ± 1.35 BCDb
	MKS	1.20 ± 0.06 Aa	44.38 ± 2.78 ABa	1.34 ± 0.04 Ba	38.82 ± 1.30 Ba

Note: Different uppercase letters indicate significant differences among different triticale varieties within the same growth stage (*p* < 0.05); different lowercase letters indicate significant differences for the same triticale variety across different growth stages (*p* < 0.05).

**Table 7 plants-14-02942-t007:** The neutral detergent fiber and acid detergent fiber of different triticale varieties in 2024 and 2025 at different growth stages.

Variety	Growth Periods	Record of 2024		Record of 2025	
		Neutral Detergent Fiber %	Acid Detergent Fiber %	Neutral Detergent Fiber %	Acid Detergent Fiber %
QSM-1	BTS	59.53 ± 2.84 ABab	31.10 ± 1.47 Dab	58.06 ± 1.58 ABb	30.94 ± 0.81 Ba
	HDS	62.93 ± 1.50 ABCa	34.50 ± 1.13 ABCa	61.23 ± 0.67 ABa	32.16 ± 1.26 Ba
	FLS	55.97 ± 1.50 ABb	31.50 ± 0.75 ABab	51.67 ± 0.32 Cc	33.76 ± 1.01 Aa
	MKS	48.17 ± 2.04 Bc	29.17 ± 3.19 Ab	49.05 ± 0.85 BCd	30.45 ± 1.16 Aa
QSM-2	BTS	58.30 ± 5.67 ABab	32.47 ± 2.73 CDa	54.40 ± 2.82 Bb	31.81 ± 0.86 Bb
	HDS	65.00 ± 3.98 ABa	35.70 ± 2.09 ABCa	62.20 ± 1.13 Aa	35.67 ± 1.35 ABa
	FLS	56.93 ± 5.42 ABab	33.50 ± 5.83 ABa	57.25 ± 2.43 ABab	30.80 ± 1.45 BCb
	MKS	53.00 ± 1.48 ABb	29.03 ± 1.81 Aa	55.90 ± 0.40 Aab	30.26 ± 0.47 Ab
QSM-3	BTS	58.80 ± 3.16 ABab	33.00 ± 1.49 BCDa	57.70 ± 1.79 ABa	32.42 ± 0.65 Ba
	HDS	60.87 ± 0.84 BCa	33.97 ± 0.31 ABCa	58.18 ± 1.93 ABa	33.54 ± 0.12 ABa
	FLS	50.27 ± 5.61 Bb	30.47 ± 2.68 Ba	53.37 ± 1.33 Ca	29.68 ± 1.19 Ca
	MKS	57.37 ± 5.84 Aab	31.63 ± 3.44 Aa	54.71 ± 2.75 ABa	31.50 ± 2.04 Aa
QSM-7	BTS	62.60 ± 0.35 Aab	36.17 ± 0.67 Aa	60.51 ± 0.41 Aa	35.67 ± 0.36 Aa
	HDS	64.23 ± 0.23 ABa	36.30 ± 1.25 ABa	60.04 ± 0.47 ABa	35.83 ± 0.37 Aa
	FLS	61.50 ± 1.48 Ab	34.83 ± 2.8 ABa	60.77 ± 0.78 Aa	32.85 ± 0.64 ABb
	MKS	51.07 ± 2.20 ABc	30.60 ± 1.35 Ab	52.63 ± 1.71 ABCb	31.10 ± 0.516 Ac
QSM-8	BTS	57.00 ± 0.89 Ba	35.30 ± 0.36 ABCa	55.49 ± 1.14 ABa	34.79 ± 0.32 Aa
	HDS	57.73 ± 4.10 Ca	33.53 ± 4.09 BCab	56.81 ± 1.74 Ba	32.97 ± 1.52 ABab
	FLS	55.63 ± 5.35 ABab	32.57 ± 1.78 ABab	57.26 ± 1.74 ABa	31.64 ± 0.29 ABCb
	MKS	47.20 ± 6.06 Bb	29.17 ± 2.89 Ab	48.54 ± 2.62 Cb	30.57 ± 0.87 Ab
JSM-2	BTS	58.33 ± 0.78 ABa	30.80 ± 2.00 Da	56.83 ± 0.89 ABa	31.17 ± 0.83 Ba
	HDS	57.97 ± 4.11 Ca	31.87 ± 2.40 Ca	57.58 ± 2.32 ABa	32.15 ± 1.26 Ba
	FLS	59.37 ± 2.48 Aa	34.03 ± 1.99 ABa	60.49 ± 0.88 Aa	32.08 ± 0.36 ABCa
	MKS	58.17 ± 3.30 Aa	33.30 ± 1.83 Aa	57.89 ± 1.85 Aa	33.21 ± 1.05 Aa
JSM-3	BTS	59.13 ± 0.90 ABb	35.80 ± 1.22 ABa	57.68 ± 0.79 ABa	35.22 ± 0.47 Aa
	HDS	67.07 ± 2.55 Aa	37.83 ± 1.87 Aa	61.99 ± 0.73 Aa	36.44 ± 0.88 Aa
	FLS	61.60 ± 1.51 Ab	36.23 ± 1.72 Aa	60.27 ± 0.09 Aa	33.59 ± 0.31 ABa
	MKS	52.30 ± 1.84 ABc	29.23 ± 1.55 Ab	57.69 ± 0.74 Aa	30.15 ± 0.61 Aa

Note: Different uppercase letters indicate significant differences among different triticale varieties within the same growth stage (*p* < 0.05); different lowercase letters indicate significant differences for the same triticale variety across different growth stages (*p* < 0.05).

**Table 8 plants-14-02942-t008:** The relative feeding value and relative forage quality of different triticale varieties in 2024 and 2025 at different growth stages.

Variety	Growth Periods	Record of 2024		Record of 2025	
		Relative Feeding Value	Relative Forage Quality	Relative Feeding Value	Relative Forage Quality
QSM-1	BTS	101.15 ± 3.54 ABb	69.91 ± 4.11 ABbc	104.01 ± 3.60 Ad	138.05 ± 6.76 ABbc
	HDS	90.79 ± 4.35 ABc	79.04 ± 2.37 Ab	97.06 ± 2.48 ABc	125.47 ± 6.16 ABc
	FLS	106.91 ± 3.47 ABb	99.28 ± 1.81 Aa	106.53 ± 1.76 ABb	134.09 ± 4.38 ABb
	MKS	128.11 ± 7.08 Aa	61.87 ± 1.93 ABbc	123.64 ± 0.43 Aa	162.71 ± 3.95 Aa
QSM-2	BTS	102.21 ± 13.21 Aab	73.69 ± 8.23 ABbc	110.30 ± 6.40 Aa	143.19 ± 10.10 Aa
	HDS	86.82 ± 8.95 Bb	77.28 ± 8.56 Ab	91.60 ± 3.17 Bb	111.31 ± 6.34 Bb
	FLS	103.41 ± 17.41 ABab	87.73 ± 2.68 ABCa	106.01 ± 6.51 ABab	141.50 ± 12.83 ABa
	MKS	116.50 ± 2.38 ABa	57.76 ± 4.57 Cc	108.71 ± 0.64 ABa	146.28 ± 1.86 ABa
QSM-3	BTS	100.14 ± 6.73 ABb	71.79 ± 4.56 Bb	102.80 ± 3.61 ABa	132.53 ± 6.82 Aa
	HDS	95.31 ± 1.72 ABb	88.67 ± 4.67 Aa	101.16 ± 3.37 ABa	126.73 ± 4.02 ABa
	FLS	121.75 ± 14.54 Aa	76.21 ± 8.72 BCab	116.06 ± 5.13 Aa	156.33 ± 9.74 Aa
	MKS	105.29 ± 14.34 Bab	66.51 ± 1.14 ABc	111.05 ± 8.70 ABa	145.30 ± 15.00 ABa
QSM-7	BTS	89.98 ± 0.78 Bb	62.55 ± 0.44 Ac	92.82 ± 0.40 Bb	114.24 ± 0.74 Cb
	HDS	87.23 ± 0.82 Bb	70.08 ± 1.36 Ab	93.29 ± 0.63 Bb	114.72 ± 1.20 ABb
	FLS	93.66 ± 5.46 Bb	97.39 ± 2.41 Aa	97.71 ± 1.45 Bb	123.56 ± 3.66 Bb
	MKS	118.80 ± 6.57 ABa	62.36 ± 3.48 ABCc	114.99 ± 4.96 ABa	150.48 ± 6.72 ABa
QSM-8	BTS	100.19 ± 1.19 ABb	76.19 ± 1.38 Aab	103.99 ± 2.45 Ab	128.14 ± 3.59 ABCb
	HDS	101.88 ± 12.50 Ab	80.91 ± 9.07 Aab	106.43 ± 5.03 Ab	133.01 ± 9.87 Ab
	FLS	107.68 ± 13.73 ABab	98.06 ± 6.48 Aa	106.80 ± 3.06 ABb	136.64 ± 4.80 ABb
	MKS	131.98 ± 20.77 Aa	74.74 ± 6.21 Ab	124.13 ± 5.64 Aa	163.67 ± 10.79 Aa
JSM-2	BTS	103.65 ± 3.50 Aa	72.70 ± 1.14 ABa	109.94 ± 0.94 Aa	139.80 ± 2.00 ABa
	HDS	103.38 ± 10.10 Aa	74.60 ± 2.38 Aa	107.65 ± 5.23 Aa	134.30 ± 9.25 Aa
	FLS	97.81 ± 5.65 Ba	73.44 ± 5.01 Ca	103.12 ± 1.80 Ba	127.13 ± 3.31 Ba
	MKS	100.90 ± 7.85 Ba	74.14 ± 6.29 ABa	104.33 ± 3.93 Ba	129.16 ± 8.21 Ba
JSM-3	BTS	95.73 ± 1.56 ABb	70.69 ± 1.29 ABab	101.45 ± 0.42 ABa	121.79 ± 0.71 BCa
	HDS	81.69 ± 4.95 Bc	74.85 ± 4.56 Aab	94.33 ± 2.22 Ba	108.93 ± 4.04 Ba
	FLS	91.65 ± 4.22 Bb	89.97 ± 3.30 ABa	100.31 ± 0.59 Ba	121.86 ± 0.93 Ba
	MKS	117.87 ± 5.88 ABa	60.12 ± 2.87 BCb	110.57 ± 1.68 ABa	142.36 ± 3.34 ABa

Note: Different uppercase letters indicate significant differences among different triticale varieties within the same growth stage (*p* < 0.05); different lowercase letters indicate significant differences for the same triticale variety across different growth stages (*p* < 0.05).

**Table 9 plants-14-02942-t009:** Eigen value, contribution rates and eigenvector of principal components of different indicators of triticale varieties in 2024 and 2025.

Indicator	Comprehensive Indicators				
	2024		2025		Ranking
	Principal Component CI_1_	Principal Component CI_2_	Principal Component CI_1_	Principal Component CI_2_	
Plant height	0.108	0.173	0.093	0.20	5
Tiller number	−0.104	0.065	−0.065	0.113	10
Stem diameter	−0.051	0.162	−0.06	0.037	13
Number of leaves	0.065	0.205	0.039	0.18	8
Fresh biomass yield	0.07	0.228	0.062	0.234	7
Dry matter yield	0.106	0.146	0.103	0.17	4
Dry matter content	0.129	0.087	0.129	−0.008	2
CA	−0.136	0.010	−0.128	−0.042	14
CP	−0.129	0.007	−0.128	−0.027	15
EE	0.027	−0.166	−0.006	−0.227	12
SS	0.139	−0.012	0.138	0.044	1
NDF	−0.106	0.143	−0.105	0.143	11
ADF	−0.096	0.195	−0.11	0.135	9
RFV	0.109	−0.153	0.121	−0.121	3
RFQ	−0.003	0.136	0.121	−0.157	6
Eigen values	6.71	3.22	6.51	3.29	
Contributive rate/%	44.66	21.44	0.43	0.22	
Cumulative contributive rate/%	44.66	66.10	43.44	65.34	

**Table 10 plants-14-02942-t010:** Comprehensive evaluation of different triticale varieties in 2024 and 2025.

Variety	Growth Periods	Euclidean Distance to Positive Ideal Solution (*D*^+^)		Euclidean Distance to Negative Ideal Solution (*D*^−^)		Closeness Coefficient (*Ci*)
QSM-1	BTS	0.72	0.61	0.41	0.57	0.421
	HDS	0.61	0.64	0.6	0.50	0.465
	FLS	0.62	0.58	0.57	0.64	0.503
	MKS	0.62	0.58	0.59	0.70	0.514
QSM-2	BTS	0.79	0.60	0.34	0.70	0.420
	HDS	0.67	0.80	0.49	0.36	0.365
	FLS	0.70	0.55	0.50	0.55	0.460
	MKS	0.6	0.66	0.62	0.55	0.480
QSM-3	BTS	0.77	0.73	0.33	0.41	0.335
	HDS	0.65	0.61	0.48	0.57	0.455
	FLS	0.63	0.56	0.56	0.71	0.515
	MKS	0.70	0.77	0.59	0.34	0.385
QSM-7	BTS	0.76	0.66	0.50	0.53	0.420
	HDS	0.68	0.72	0.59	0.36	0.400
	FLS	0.55	0.76	0.62	0.47	0.465
	MKS	0.64	0.48	0.48	0.59	0.490
QSM-8	BTS	0.55	0.59	0.74	0.60	0.540
	HDS	0.44	0.40	0.77	0.77	0.650
	FLS	0.44	0.46	0.82	0.63	0.615
	MKS	0.29	0.56	0.84	0.71	0.650
JSM-2	BTS	0.72	0.65	0.49	0.61	0.445
	HDS	0.63	0.59	0.62	0.66	0.515
	FLS	0.55	0.74	0.65	0.43	0.455
	MKS	0.73	0.73	0.52	0.44	0.400
JSM-3	BTS	0.59	0.63	0.67	0.64	0.515
	HDS	0.66	0.68	0.65	0.63	0.490
	FLS	0.48	0.67	0.72	0.56	0.525
	MKS	0.62	0.56	0.67	0.71	0.540

**Table 11 plants-14-02942-t011:** The physical and chemical properties of the soil.

Nutrients Parameters	Value
PH value	8.70
Electrical conductivity	2.61 ms·cm^−1^
Total nitrogen	0.83 g·kg^−1^
Total phosphorus	0.71 g·kg^−1^
Available potassium	135.00 mg·kg^−1^
Available phosphorus	29.08 mg·kg^−1^
Nitrate nitrogen	21.87 mg·kg^−1^
Organic matter	10.76 g·kg^−1^
Total dissolved solids	11.00 g·kg^−1^

**Table 12 plants-14-02942-t012:** The varieties of triticale tested and their basic information.

Sample Number	Variety Name	Variety Type	Variety Source	Breeding Sites
QSM-1	Qingsimai No. 1	Spring-type medium-maturing forage variety	Academy of Animal Science and Veterinary Medicine, Qinghai University	Qinghai Province
QSM-2	Qingsimai No. 2	Spring-type medium maturing forage variety	Academy of Animal Science and Veterinary Medicine, Qinghai University	Qinghai Province
QSM-3	Qingsimai No. 3	Spring-type medium maturing forage variety	Academy of Animal Science and Veterinary Medicine, Qinghai University	Qinghai Province
QSM-7	Qingsimai No. 7	Spring-type medium-late maturing forage variety	Academy of Animal Science and Veterinary Medicine, Qinghai University	Qinghai Province
QSM-8	Qingsimai No. 8	Spring-type medium-late maturing forage variety	Academy of Animal Science and Veterinary Medicine, Qinghai University	Qinghai Province
JSM-2	Jisimai No. 2	Spring-type late-maturing forage variety	Ningxia Xibei Agriculture, Forestry and Animal Husbandry Ecological Technology Co., Ltd.	Hebei Province
JSM-3	Jisimai No. 3	Spring-type late-maturing forage variety	Ningxia Xibei Agriculture, Forestry and Animal Husbandry Ecological Technology Co., Ltd.	Hebei Province

## Data Availability

The original contributions presented in this study are included in the article. Further inquiries can be directed to the corresponding author.

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
