# Peer review of "The Impact of Varieties and Growth Stages on the Production Performance and Nutritional Quality of Forage Triticale in the Qaidam Basin"

_plants, 2025, doi:10.3390/plants14192942_

Round 1

Reviewer 1 Report (Previous Reviewer 3)

Comments and Suggestions for Authors

Main contributions of the manuscript – The study evaluates the suitability of triticale varieties for a specific cultivation region based on 15 indicators. The manuscript has been improved, most importantly by including the analysis of data from two study years. Therefore, overall, I consider the article suitable for publication with minor revisions.

However, there are still some minor suggestions for improving the manuscript: 

Suggestions for improvement

1) I recommend moving Table 1 to the supplementary material. In addition, the table title should be clarified, and the years should be specified more precisely.

2) Methodology – After reading the methods section, it remains unclear how the phenological stages were calculated. Please clarify whether they represent the number of days from sowing to each developmental stage, or the number of days between successive stages.

3) I recommend supplementing the methodology by specifying the seeding rate used.

4) Quite a number of technical errors were noticed, which should be carefully revised.

Author Response

Reviewer 2 Report (Previous Reviewer 1)

Comments and Suggestions for Authors

I agree to publish, but check lines 90-91 for the lack of the PART 2 TITLE. Results? Best regards  

Author Response

This manuscript is a resubmission of an earlier submission. The following is a list of the peer review reports and author responses from that submission.

Round 1

Reviewer 1 Report

Comments and Suggestions for Authors

The article is interesting and well presented. The writing is adequate for the most part. There were some inaccuracies that should be corrected, but I recommend that it be published. Here are some corrections and other suggestions

REVIEW – sugestions to correct, at least review or rethink!

The Impact of Different Varieties and Growth Stages on the  Production Performance and Nutritional Quality of Forage Triticale in the Qaidam Basin

About title: sugest eliminate ‘different’

The Impact of Varieties and Growth Stages on the  Production Performance and Nutritional Quality of Forage Triticale in the Qaidam Basin

Sugestions to correct or review:

p.1 l.35-36 – What’s mean “Strong sunlight and significant diurnal temperature variations”? I’m from Southern Brazil – have no idea about it – You should put some information...

p.2 - Table 1. The growth stage of different triticale varieties (to eliminte diferente)

p.2 l.105-106 – the plant height of JSM2 and QSM7 are not diferente on milking and mature stages

p.2. l.107 to 111 – impossible to analyse because there is no on Figure 1 B (tillers number). Please review and correct.  Figure 1b and 1d are the same

Figure 1. Agronomic traits and grass yield of the participating triticale varieties at different growth 135 periods. (A) Plant height of the triticale variety; (B) Tillering number of the triticale variety (TO INCLUDE) ; (C) Stem 136 thickness of the triticale variety; (D) Number of green leaves of the triticale variety; (E) Fresh grass 137 yield of the triticale variety; (F) Dry grass yield of the triticale variety

p.3 l.117-118  There is was no significant difference – I think there is... p<0.0001 – see table  2 – In fact there was no interaction between variety x growth stage

p.3 l. 125 - Each variety reached its maximum yield at the flowering stage, followed by a decrease at the milk stage.  I think should be “trended”

p.4 l.136 to Correct on figure 1 E – varieties instead if varirties

p.7 l.216-222 review order of top five

p.10 l.308  -  Is it trirye and triticale synonyms?  Keep triticale along total article

  1. 10 l. 315 – review triplet rye?
  2. Materials and Methods

   4.1. General Situations of Experimental Site

p.11 l. 349... 4.2 Plant Materials  - I think Table 6 should be repalced by short text

p.12 l. 350 – repeated order 4.2 Experimental Design (should be 4.3...?!)

p.12 l. 378 – A new paragraph for stalk diameter

Author Response

Point-by-point response to the reviewer comments

Dear Reviewers,

Thank you for your thoughtful suggestions and insights, which have benefited from the manuscript. I am looking forward to working with you to move this manuscript closer to publication in “Plants”.

The manuscript has been rechecked and the necessary changes have been made in accordance with your suggestions. The responses to all comments have been prepared and attached below. We have tried our best to solve the problems you proposed, and we hope that the revised manuscript is now suitable for publication in the journal “Plants”. If you have any questions remained about this paper, please feel free to contact us.

Reviwer #1:

About title: suggest eliminate ‘different’

Re: Thank you very much for your valuable comments on this study. As suggested, we have revised the original title by removing the word "different". The updated title is now: The Impact of Varieties and Growth Stages on the Production Performance and Nutritional Quality of Forage Triticale in the Qaidam Basin.

Suggestions to correct or review:

35-36–What’s mean “Strong sunlight and significant diurnal temperature variations”? I’m from Southern Brazil–have no idea about it–You should put some information…

Re: Thank you very much for your time and effort in reviewing our manuscript. We have already made the revisions in the manuscript. (line36-line39). In response to your feedback, we further elaborated and explained the original content and cited some references as the basis. We have made the following revisions:This region is characterized by an arid and frigid climatic regime, dry and rarefied air, pronounced diurnal temperature fluctuations between morning and evening, and intense solar radiation—with an average solar radiation flux exceeding 6000 joules per square meter per day.

Table 1. The growth stage of different triticale varieties (to eliminate different)

Re: We would like to express our sincere gratitude to the reviewers for their careful reading and constructive comments on our manuscript. These comments have been extremely helpful in improving the quality of our paper. According to the suggestion, we have deleted the "different" in the title.

105-106–the plant height of JSM2 and QSM7 are not different on milking and mature stages

Re: Thank you very much for your valuable comments. We have already made the revisions in the manuscript. (line 128).

107-111-impossible to analyse because there is no on Figure 1 B (tillers number). Please review and correct. Figure 1b and 1d are the same.

Figure 1. Agronomic traits and grass yield of the participating triticale varieties at different growth 135 periods. (A) Plant height of the triticale variety; (B) Tillering number of the triticale variety (TO INCLUDE); (C) Stem 136 thickness of the triticale variety; (D) Number of green leaves of the triticale variety; (E) Fresh grass 137 yield of the triticale variety; (F) Dry grass yield of the triticale variety

Re: Thank you very much for pointing out this mistake. We have revised the original photograph in the manuscript. Thank you for your reminder. (line 168).

117-118–There is was no significant difference – I think there is... p<0.0001 – see table 2 – In fact there was no interaction between variety x growth stage

Re: Thank you for your review. We have revised the relevant contents of the manuscript and provided explanations in the table notes (line 114- line 119). We consider that in the significance analysis, P>0.05 is considered insignificant, 0.01<P<0.05 is a significant difference, and P≤0.001 is extremely significant. In the SPSS software calculation, (P<0.001) is displayed as P=0, usually because the value is less than the calculation accuracy, and it is defaulted to be in this range.

125-Each variety reached its maximum yield at the flowering stage, followed by a decrease at the milk stage. I think should be “trended”

Re: Thank you for your review. We have revised the relevant content to the manuscript. (line 147)

136 to Correct on figure 1 E –varieties instead if varirties.

Re: Thank you very much for pointing out the word error in our manuscript. We have revised the original photograph in the manuscript. We truly appreciate your attention to detail and valuable feedback.

216-222 review order of top five

Re: Your comments are highly appreciated. We have already made the revisions in the manuscript. (line 289- line 290)

308 - Is it trirye and triticale synonyms? Keep triticale along total article

Re: Thank you for your reminder. This issue has been revised throughout the full text.

315–review triplet rye?

Re: Thank you for your review. We have made detailed revisions to the relevant content in the manuscript and are very grateful for your attention to the details and valuable feedback.

  1. Materials and Methods

4.1. General Situations of Experimental Site

Re: Thank you for pointing out the mistakes in the words. We have corrected the original manuscript. (line 420)

4.2 Plant Materials - I think Table 6 should be repalced by short text

Re: Thank you very much for your valuable comments. We have already made the revisions in the manuscript. (line 449).

350 – repeated order 4.2 Experimental Design (should be 4.3...?!)

Re: Thank you very much for pointing out this mistake. We have already made the correction in the manuscript. (line 450)

378 – A new paragraph for stalk diameter

Re: Thank you very much for pointing out this mistake. We have already made the correction in the manuscript. We truly appreciate your attention to detail and valuable feedback. (line 480)

Reviewer 2 Report

Comments and Suggestions for Authors

Dear Authors,

The topic is both important and timely. The shortage of forage in high-altitude and arid regions like the Qaidam Basin indeed poses a significant challenge to livestock development. The authors designed a field experiment involving seven triticale varieties and analyzed their yield and forage quality at four different growth stages. The research idea is relevant and well aligned with the needs of agriculture under harsh environmental conditions. However, the manuscript requires substantial revisions before it is ready for publication.

Main comments:

The introduction does not clearly state the hypothesis or precise research objective. Did the authors expect significant differences between varieties and growth stages in terms of yield and quality? It would be helpful to include this explicitly, preferably at the end of the introduction, so the reader understands what exactly was tested.

The terminology used by the authors is imprecise. The terms like “grass yield” and “hay yield” are not ideal. Instead of “grass yield,” it’s better to use “forage yield” or “fresh biomass yield.” Similarly, “dry matter yield” is a more accurate term than “hay yield.”

The paper reports both fresh biomass and dry matter yields but does not calculate the dry matter content (%). Dry matter content is a key indicator of silage suitability, so I suggest including average dry matter content for each growth stage.

The authors use the RFV index to assess forage quality. It should be noted that RFV is outdated; the Relative Forage Quality (RFQ) index is more commonly used today. RFQ better reflects triticale quality because it accounts for fiber digestibility variability, which RFV does not. Additionally, RFQ more accurately represents the energy value and real digestibility of the forage, which is crucial when planning ruminant diets. If the authors lack data to calculate RFQ, they should at least acknowledge that RFV is a simplified index with limitations.

All data come from a single season and location. This is a significant limitation, especially in a region with high climate variability. The authors should explicitly mention this in the discussion or conclusions and emphasize that the results need to be confirmed in subsequent years.

Morphological traits like leaf and stem number were measured, but the authors did not explain why these are important from a forage quality perspective. This should be clarified. A correlation analysis between quality traits and yield, e.g., whether higher dry matter yield is linked to lower protein or higher NDF, would add value.

The paper lacks any reflection on the limitations of the methods used or suggestions for future research. This is important. For example, how do the results translate to actual animal nutrition? Should additional forage quality indicators be included? Adding such considerations to the discussion or conclusion section would strengthen the manuscript.

Finally, the style and language need thorough proofreading. There are numerous language errors (e.g., “triplet rye” instead of triticale). I recommend a language edit by a professional editor or native speaker.

Author Response

Point-by-point response to the reviewer comments

Dear Reviewers,

Thank you for your thoughtful suggestions and insights, which have benefited from the manuscript. I am looking forward to working with you to move this manuscript closer to publication in “Plants”.

The manuscript has been rechecked and the necessary changes have been made in accordance with your suggestions. The responses to all comments have been prepared and attached below. We have tried our best to solve the problems you proposed, and we hope that the revised manuscript is now suitable for publication in the journal “Plants”. If you have any questions remained about this paper, please feel free to contact us.

Reviwer #2

The topic is both important and timely. The shortage of forage in high-altitude and arid regions like the Qaidam Basin indeed poses a significant challenge to livestock development. The authors designed a field experiment involving seven triticale varieties and analyzed their yield and forage quality at four different growth stages. The research idea is relevant and well aligned with the needs of agriculture under harsh environmental conditions. However, the manuscript requires substantial revisions before it is ready for publication.

Main comments:

The introduction does not clearly state the hypothesis or precise research objective. Did the authors expect significant differences between varieties and growth stages in terms of yield and quality? It would be helpful to include this explicitly, preferably at the end of the introduction, so the reader understands what exactly was tested.

Re: We would like to express our sincere gratitude to the reviewers for their careful reading and constructive comments on our manuscript. According to your suggestion, we have added the elaboration of relevant content in the introduction part. (line78- line 84)

The terminology used by the authors is imprecise. The terms like “grass yield” and “hay yield” are not ideal. Instead of “grass yield,” it’s better to use “forage yield” or “fresh biomass yield.”  Similarly, “dry matter yield” is a more accurate term than “hay yield.”

Re: Thank you for your rigorous scrutiny of terminology precision. We fully agree with your suggestion and have corrected the relevant content of the full text.

The paper reports both fresh biomass and dry matter yields but does not calculate the dry matter content (%). Dry matter content is a key indicator of silage suitability, so I suggest including average dry matter content for each growth stage.

Re: Thank you for your review. We think this is an excellent suggestion. We have added the content about dry matter content in the manuscript. (line158- line164)

The authors use the RFV index to assess forage quality. It should be noted that RFV is outdated; the Relative Forage Quality (RFQ) index is more commonly used today. RFQ better reflects triticale quality because it accounts for fiber digestibility variability, which RFV does not. Additionally, RFQ more accurately represents the energy value and real digestibility of the forage, which is crucial when planning ruminant diets. If the authors lack data to calculate RFQ, they should at least acknowledge that RFV is a simplified index with limitations.

Re: Thank you for suggesting the inclusion of RFQ. We appreciate its complementary value to RFV. We have added RFQ analysis in Section 2.3, integrating both indices to evaluate forage quality comprehensively. This dual-metric approach enhances the study’s rigor by addressing yield (RFV) and nutritional parameters (RFQ). (line216- line 229)

All data come from a single season and location. This is a significant limitation, especially in a region with high climate variability. The authors should explicitly mention this in the discussion or conclusions and emphasize that the results need to be confirmed in subsequent years.

Re: We sincerely appreciate the valuable comments. We have added your suggestions in the conclusion section of the manuscript. These comments have been extremely helpful in improving the quality of our paper. (line534- line 539)

Morphological traits like leaf and stem number were measured, but the authors did not explain why these are important from a forage quality perspective. This should be clarified. A correlation analysis between quality traits and yield, e.g., whether higher dry matter yield is linked to lower protein or higher NDF, would add value.

Re: Thank you very much for your valuable comments. These comments have been extremely helpful in improving the quality of our paper. According to the suggestion, we have already supplemented the relevant content in the manuscript. (line329- line 332)

The paper lacks any reflection on the limitations of the methods used or suggestions for future research. This is important. For example, how do the results translate to actual animal nutrition? Should additional forage quality indicators be included? Adding such considerations to the discussion or conclusion section would strengthen the manuscript.

Re: Thank you very much for your valuable comments. We have added future suggestions in the discussion and results section of the paper. (line348- line 352)

Finally, the style and language need thorough proofreading. There are numerous language errors (e.g., “triplet rye” instead of triticale). I recommend a language edit by a professional editor or native speaker.

Re: We feel great thanks for your professional review work on our article. We have checked the entire text and made the necessary revisions. And we hope the revised manuscript could be acceptable for you.

Reviewer 3 Report

Comments and Suggestions for Authors

The paper focuses on evaluating local triticale breeding lines, determining their suitability for forage cultivation, and selecting the ideal harvesting stage. The paper focuses on selecting the best cultivars for a certain region in China. It provides clear applied science examples that growers can apply in production.

Generally Materials and methods are well described and repeatable; nonetheless, some inaccuracies are likely to be identified.

  • The authors mention the overall climatic conditions in the region, but they do not specify the specific characteristics of the specific vegetation period, which, of course, influences the growth of cultivars.
  • It would have been advised to include a control variety in the comparisons, as it is not possible to determine in general how well triticale adapts to growth.
  • Type of triticale varieties should be mentioned, such as spring or winter type.
  • Positive and negative ideal solutions, used for TOPSIS model should be included.

Discussion – rows 232-238 the authors refer to studies on winter triticale (and also rye), although this is not appropriate for the subject of this research. Furthermore, the authors failed to look into the meteorological conditions of the specific growing season, which could have influenced the study's findings.

  • Row 282-283 – authors recommend three varieties of triticale for cultivation, although It is unclear how these varieties are recommended, Are recommendations based only on yield results? If so, why are other parameters evaluated?

Author Response

Point-by-point response to the reviewer comments

Dear Reviewers,

Thank you for your thoughtful suggestions and insights, which have benefited from the manuscript. I am looking forward to working with you to move this manuscript closer to publication in “Plants”.

The manuscript has been rechecked and the necessary changes have been made in accordance with your suggestions. The responses to all comments have been prepared and attached below. We have tried our best to solve the problems you proposed, and we hope that the revised manuscript is now suitable for publication in the journal “Plants”. If you have any questions remained about this paper, please feel free to contact us.

Reviwer #3

A brief summary (one short paragraph) outlining the aim of the paper, its main contributions and strengths.

The paper focuses on evaluating local triticale breeding lines, determining their suitability for forage cultivation, and selecting the ideal harvesting stage. The paper focuses on selecting the best cultivars for a certain region in China. It provides clear applied science examples that growers can apply in production.

Article:

Generally Materials and methods are well described and repeatable; nonetheless, some inaccuracies are likely to be identified.

The authors mention the overall climatic conditions in the region, but they do not specify the specific characteristics of the specific vegetation period, which, of course, influences the growth of cultivars.

Re: Thank you for your review. We think this is an excellent suggestion. According to your suggestion, we have added relevant content in the Materials and methods part. (line 443)

It would have been advised to include a control variety in the comparisons, as it is not possible to determine in general how well triticale adapts to growth.

Re: Thank you for your thoughtful suggestion to include a control cultivar. We appreciate the importance of control variety for adaptability assessment. Notably, our study focuses on relative adaptability evaluation among seven triticale varieties, using mutual comparisons as the primary analytical framework. We propose that future multi-location studies (including the control) could complement these findings. We have added relevant suggestions in the results section of the paper.

Type of triticale varieties should be mentioned, such as spring or winter type.

Re: Thank you very much for your review. We highly agree with your suggestions. We have supplemented the types of triticale in the table. (line 449)

Positive and negative ideal solutions, used for TOPSIS model should be included.

Re: Thank you very much for your valuable comments. We have already supplemented the relevant content in the manuscript.

Discussion-rows 232-238 the authors refer to studies on winter triticale (and also rye), although this is not appropriate for the subject of this research. Furthermore, the authors failed to look into the meteorological conditions of the specific growing season, which could have influenced the study's findings.

Re: Thank you very much for your valuable comments. We have already made the revisions in the manuscript, and climatic conditions have been added to the General Situations of the Experimental Site. ((line314 and line 443)

Row 282-283 – authors recommend three varieties of triticale for cultivation, although It is unclear how these varieties are recommended, are recomendations based only on yiled results? If so, why are other parameters evaluated?

Re: Thank you very much for pointing out this mistake. This part of the content is about three small rye varieties recommended based on forage yield. We think it lacks consideration of the combination of yield and quality. After some thought, this part of the content has been deleted. Thank you very much for your review. (line371- line 372)

Specific comments referring to line numbers, tables or figures that point out inaccuracies within the text or sentences that are unclear.

Table 1, I would propose using the number of days from sowing to the phenological phase rather than month and day. Also, a suggestion to the authors: perhaps Tables 1 and 3 should be included in the supplementary files.

Re: We sincerely appreciate the valuable comments. According to your suggestion, the months and day in the table have been changed to the number of days from sowing to the phenological period. According to the suggestions, Table 1 has been taken as supplementary material. As Table 3 requires explanation and elaboration of the content of the paper, I think it can be retained in the original text. Should journal formatting necessitate adjustments, we are happy to further optimize table layouts. (Table S1)

Figure 3- replace Number of leaves /pieces on Number of leaves /plant.

Re: We would like to express our sincere gratitude to the reviewers for their careful reading and constructive comments on our manuscript. We have made detailed revisions to the relevant content in the manuscript and are very grateful for your attention to the details and valuable feedback. (Figure 3)

In different parts of the article, the authors note different varieties as the best – in Conclusions JSM-3 and QSM-8, in Abstract QSM-8 > JSM-3 > QSM-7 > JSM-2 > QSM-1. It is necessary to harmonize and more clearly show which varieties are the best and why

Re: Thank you very much for pointing out this mistake. We have checked the entire text and made the necessary revisions. And we hope the revised manuscript could be acceptable for you. (line290)

Round 2

Reviewer 2 Report

Comments and Suggestions for Authors

Dear Authors,

Thank you for submitting the revised version of your manuscript and for addressing my comments in a thorough and constructive manner. I appreciate the improvements made to the manuscript, and I am satisfied with your responses and the revisions provided. From my perspective, the paper is now suitable for publication.

Reviewer 3 Report

Comments and Suggestions for Authors

Although significant improvements have been made to the article to increase its quality, it still has one fundamental weakness: it presents the findings of only one year of research. I believe that in order for the study to be published in the journal Plants, the research should be continued in 2025 and the article resubmitted, summarising the outcomes of the two preceding years. 

Author Response

Thank you for your critical and helpful comment, please see the attachment.
